# Effective and Efficient Masked Image Generation Models

Zebin You [1 2 3 †]   Jingyang Ou [1 2 3]   Xiaolu Zhang [4]   Jun Hu [4]   JUN ZHOU [4]   Chongxuan Li [1 2 3 ¶]

## Abstract

Although masked image generation models and masked diffusion models are designed with different motivations and objectives, we observe that they can be unified within a single framework. Building upon this insight, we carefully explore the design space of training and sampling, identifying key factors that contribute to both performance and efficiency. Based on the improvements observed during this exploration, we develop our model, referred to as **eMIGM**. Empirically, eMIGM demonstrates strong performance on ImageNet generation, as measured by Fréchet Inception Distance (FID). In particular, on ImageNet $256 \times 256$, with similar number of function evaluations (NFEs) and model parameters, eMIGM outperforms the seminal VAR. Moreover, as NFE and model parameters increase, eMIGM achieves performance comparable to the state-of-the-art continuous diffusion model REPA while requiring less than 45% of the NFE. Additionally, on ImageNet $512 \times 512$, eMIGM outperforms the strong continuous diffusion model EDM2. Code is available at https://github.com/ML-GSAI/eMIGM.

## 1. Introduction

Masked modeling has proven effective across various domains, including self-supervised learning (He et al., 2022a; Bao et al., 2021; Devlin, 2018), label to image generation (Li et al., 2023; Chang et al., 2022; Li et al., 2024; Ni et al., 2024), text to image generation (Bai et al., 2024; Shao et al., 2024) and text generation (Sahoo et al., 2024;

Shi et al., 2024; Lou et al., 2024a). In image generation, MaskGIT (Chang et al., 2022) introduced masked image generation, offering efficiency and quality improvements over autoregressive models but still lagging behind diffusion models (Ho et al., 2020; Sohl-Dickstein et al., 2015; Song et al., 2020) due to information loss from discrete tokenization (Esser et al., 2021; Van Den Oord et al., 2017). MAR (Li et al., 2024) eliminated this bottleneck via diffusion loss, achieving strong results, yet key factors (e.g., masking schedule, loss function) remain underexplored. Moreover, with limited sampling steps (e.g., 16), its performance falls short of coarse-to-fine next-scale prediction model VAR (Tian et al., 2024).

In parallel, masked diffusion models (Sahoo et al., 2024; Shi et al., 2024; Lou et al., 2024a; Ou et al., 2024) have shown promise in text generation, demonstrating scaling properties (Nie et al., 2024) similar to ARMs and offering a principled probabilistic framework for training and inference. However, their applicability to image generation remains an open question.

We propose a unified framework integrating masked image modeling (Chang et al., 2022; Li et al., 2024; Bai et al., 2024) and masked diffusion models (Lou et al., 2024a; Sahoo et al., 2024; Shi et al., 2024), leveraging the strengths of both paradigms. This enables a systematic exploration of training and sampling strategies to optimize performance. For training, we find that images, due to their high redundancy, benefit from a higher masking ratio, a simple weighting function inspired by MaskGIT and MAE (He et al., 2022a) tricks, improving generation quality. We also present CFG with Mask, replacing the fake class token with a mask token for unconditional generation, further enhancing performance. For sampling, predicting fewer tokens in early stages improves results. However, early-stage guidance decreases variance, raising FID. To counter this, we propose a time interval strategy for classifier-free guidance in masked image generation, applying guidance only in later stages. This maintains strong performance while significantly accelerating sampling by reducing NFEs.

Building on our training and sampling improvements, we develop eMIGM and evaluate it on ImageNet (Deng et al., 2009) at $256 \times 256$ and $512 \times 512$ resolutions. As model parameters scale, eMIGM achieves progressively higher sam-

---

†Work done during an internship at Ant Group  ¹Gaoling School of Artificial Intelligence, Renmin University of China, Beijing, China. ²Beijing Key Laboratory of Research on Large Models and Intelligent Governance. ³Engineering Research Center of Next-Generation Intelligent Search and Recommendation, MOE. ⁴Ant Group. Correspondence to: Chongxuan Li <chongxuanli@ruc.edu.cn>.

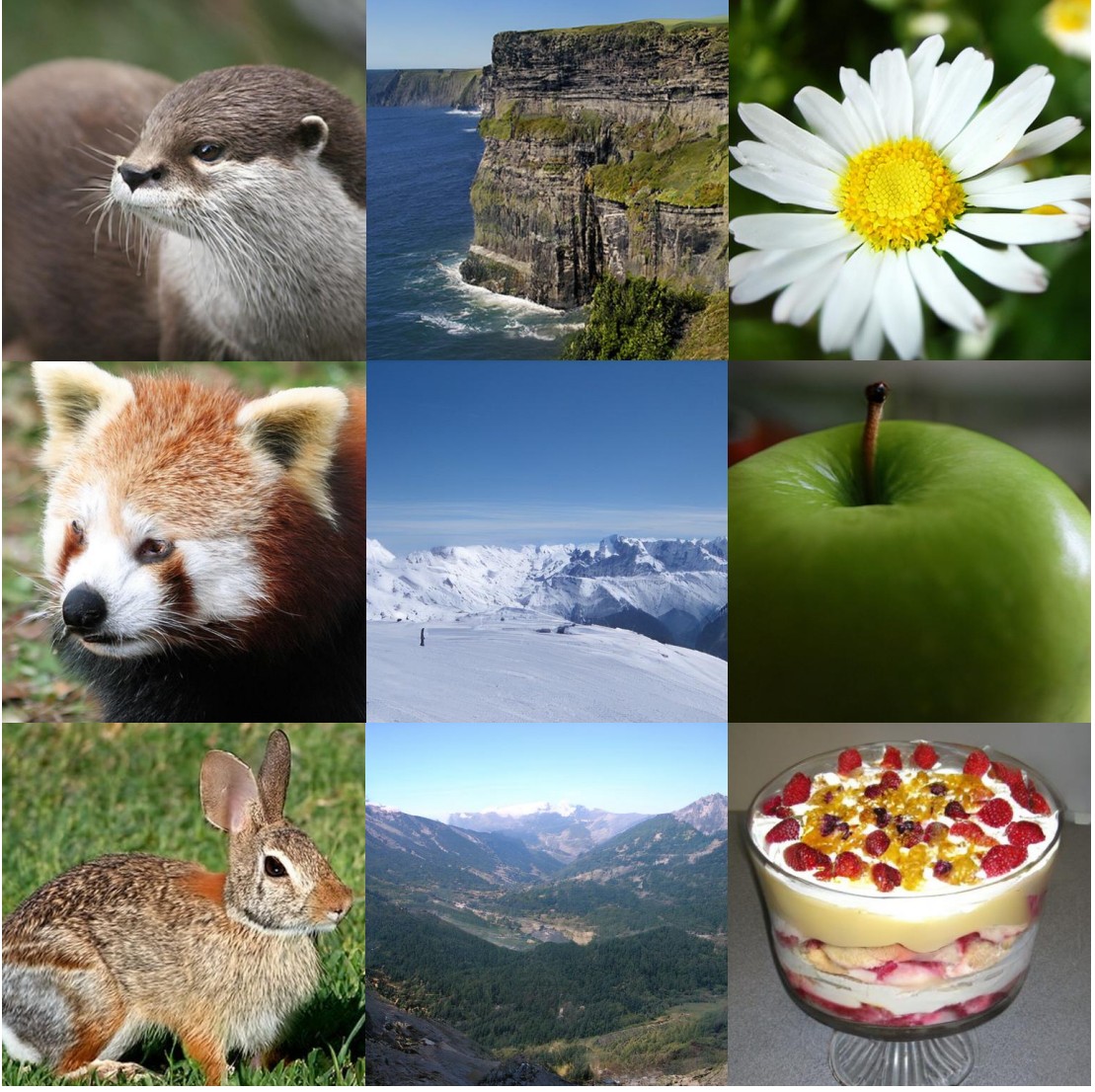

*Figure 1.* Generated samples from eMIGM trained on ImageNet $512 \times 512$.

ple quality in a predictable manner (Fig. 4(a)). Larger models further enhance efficiency, maintaining superior quality with similar training FLOPs and sampling time (Fig.4(b), Fig. 4(c)). Notably, eMIGM delivers high-quality samples with few sampling steps. On ImageNet $256 \times 256$, with similar NFEs and model parameters, it consistently outperforms VAR (Tian et al., 2024). Increasing NFE and model size, our best-performing eMIGM-H becomes comparable to state-of-the-art diffusion models like REPA (Yu et al., 2024) (FID 1.57 vs. 1.42)—without requiring self-supervised features. On ImageNet $512 \times 512$, eMIGM-L surpasses EDM2 (Karras et al., 2024) while using a lower parameter count, demonstrating efficiency and scalability. Qualitatively, eMIGM generates realistic and diverse images (Fig. 1).

In summary, our key contributions are as follows:

- We propose a unified formulation to systematically explore the design space of masked image generation models, uncovering the role of each component.

- We introduce the time interval strategy for classifier-free guidance, maintaining high performance while significantly reducing sampling time.

- We surpass the seminar diffusion models on ImageNet $512 \times 512$.

- We demonstrate that eMIGM benefits from scaling, with larger eMIGM models achieving greater efficiency.

## 2. Preliminaries

### 2.1. Masked Image Generation

Let $\boldsymbol{x} = [\boldsymbol{x}^i]_{i=1}^N$ represent the discrete tokens of an image obtained via a VQ encoder (Esser et al., 2021; Van Den Oord et al., 2017), and let [M] denote the special mask token. We consider two seminal masked image generation methods.

**MaskGIT** (Chang et al., 2022) first extends the concept of masked language modeling from BERT (Devlin, 2018) (i.e., predicting masked tokens based on unmasked tokens) to image generation, achieving excellent performance with low sampling cost (approximately 10 sampling steps) on ImageNet (Deng et al., 2009). However, its performance degrades when the number of sampling steps increases under its default mask schedule.

During training, MaskGIT optimizes the cross entropy loss as follows. A ratio $r$ is sampled from $[0, 1]$, and based on the mask scheduling function $\gamma_r$, masked image $\boldsymbol{x}_{\overline{\mathbf{M}}}$ is sampled from masking distribution $q_{\gamma_r}(\boldsymbol{x}_{\overline{\mathbf{M}}}|\boldsymbol{x})$ that randomly masks $\lceil N\gamma_r \rceil$ tokens of $\boldsymbol{x}$ as [M].

The loss function is then defined as:

$$\mathcal{L}(\boldsymbol{x}) = \mathbb{E}_{r \sim U[0,1]} \mathbb{E}_{q_{\gamma_r}(\boldsymbol{x}_{\overline{\mathbf{M}}}|\boldsymbol{x})} \left[ \sum_{\{i|\boldsymbol{x}^i=[\mathrm{M}]\}} -\log p_{\boldsymbol{\theta}}\left(\boldsymbol{x}^i \mid \boldsymbol{x}_{\overline{\mathbf{M}}}\right) \right] . \tag{1}$$

During sampling, MaskGIT starts with an image where all tokens are masked, $\boldsymbol{x}_0$. For each iteration $t \in \{1, 2, \ldots, T\}$, the number of masked tokens is $n_t = \lceil \gamma_{\frac{t}{T}} N \rceil$, and the model receives input $\boldsymbol{x}_{\frac{t-1}{T}}$. The model predicts the probabilities for all tokens, and the $\hat{n}_t = n_{t-1} - n_t$ tokens with the highest confidence are unmasked, updating to $\boldsymbol{x}_{\frac{t}{T}}$.

**MAR** (Li et al., 2024) proposes using a diffusion model (Sohl-Dickstein et al., 2015) to model the per-token distribution, which eliminates the need for discrete tokenizers. By avoiding the information loss of discrete tokenizers, MAR achieves excellent image generation performance.

During training, MAR samples the masking ratio $m_r$ from a truncated Gaussian distribution with mean 1.0, standard deviation 0.25, truncated to $[0.7, 1.0]$. For sampling, MAR adopts a decoding strategy similar to that of MaskGIT.

### 2.2. Masked Diffusion Models

Let $\boldsymbol{x} = [\boldsymbol{x}^i]_{i=1}^N$ represent the discrete text tokens of a sentence, [M] denote the special mask token, and $\gamma_t$ represent the mask schedule. MDMs (Lou et al., 2024b; Shi et al., 2024; Sahoo et al., 2024) gradually add masks to the data in the forward process and remove them during the reverse process. Here, we focus on the parameterized form of RADD (Ou et al., 2024). Given a noise level $t \in [0, 1]$,

the forward process of MDM is defined as adding noise independently in each dimension:

$$q_{t|0}(\boldsymbol{x}_t|\boldsymbol{x}_0) = \prod_{i=0}^{N-1} q_{t|0}(\boldsymbol{x}_t^i|\boldsymbol{x}_0^i), \tag{2}$$

where

$$q_{t|0}(\boldsymbol{x}_t^i|\boldsymbol{x}_0^i) = \begin{cases} 1 - \gamma_t, & \boldsymbol{x}_t^i = \boldsymbol{x}_0^i, \\ \gamma_t, & \boldsymbol{x}_t^i = [\mathrm{M}]. \end{cases} \tag{3}$$

The training objective of MDM is to optimize the upper bound of the negative log-likelihood of the masked tokens, which defined as:

$$\mathcal{L}(\boldsymbol{x}_0) = \int_0^1 \frac{\gamma_t'}{\gamma_t} \mathbb{E}_{q(\boldsymbol{x}_t|\boldsymbol{x}_0)} \left[ \sum_{\{i|\boldsymbol{x}_t^i=[\mathrm{M}]\}} -\log p_{\boldsymbol{\theta}}(\boldsymbol{x}_0^i|\boldsymbol{x}_t) \right] dt. \tag{4}$$

Interestingly, the explicit time input of MDM is theoretically redundant [1] (Ou et al., 2024), and has also been empirically validated in image generation (Hu & Ommer, 2024).

During sampling, given two noise levels $s$ and $t$, where $0 \le s < t \le 1$, the reverse process is characterized as:

$$q_{s|t}(\boldsymbol{x}_s|\boldsymbol{x}_t) = \prod_{i=0}^{N-1} q_{s|t}(\boldsymbol{x}_s^i|\boldsymbol{x}_t), \tag{5}$$

where

$$q_{s|t}(\boldsymbol{x}_s^i|\boldsymbol{x}_t) = \begin{cases} 1, & \boldsymbol{x}_s^i = \boldsymbol{x}_t^i, \ \boldsymbol{x}_t^i \neq [\mathrm{M}], \\ \frac{\gamma_s}{\gamma_t}, & \boldsymbol{x}_s^i = [\mathrm{M}], \ \boldsymbol{x}_t^i = [\mathrm{M}], \\ \frac{\gamma_t - \gamma_s}{\gamma_t} q_{0|t}(\boldsymbol{x}_s^i|\boldsymbol{x}_t), & \boldsymbol{x}_s^i \neq [\mathrm{M}], \ \boldsymbol{x}_t^i = [\mathrm{M}], \\ 0, & \text{otherwise.} \end{cases} \tag{6}$$

## 3. Unifying Masked Image Generation

After removing the explicit time input from MDM, we observe that the MaskGIT objective (Eq. 1) can be expressed in terms of the general MDM loss formulation (Eq. 4). Specifically, the Monte Carlo expectation over $r$ in Eq. 1 is equivalent to integrating over $t$ from 0 to 1, where $r$ can be interpreted as a scaled time variable $t$ corresponding to the masking schedule. In this reinterpretation, the masked image $\boldsymbol{x}_{\overline{\mathbf{M}}}$ in MaskGIT can be understood as $\boldsymbol{x}_t$ in the general

---

[1]Unlike continuous state diffusion which require both $\boldsymbol{x}_t$ and $t$ as inputs to the model input to denoise, the mask discrete diffusion operates by using $p_{\boldsymbol{\theta}}(\boldsymbol{x}_0^i|\boldsymbol{x}_t)$ instead of $p_{\boldsymbol{\theta}}(\boldsymbol{x}_0^i|\boldsymbol{x}_t, t)$. That's because the timestep dependence can be extracted as a weight coefficient outside of the cross-entropy loss.

*Table 1.* **Comparison of different masked image modeling approaches through a unified framework.** The differences among these approaches are defined by the choice of masking distribution $q(\boldsymbol{x}_t|\boldsymbol{x}_0)$, weighting function $w(t)$, and conditional distribution $p_{\boldsymbol{\theta}}(\boldsymbol{x}_0^i \mid \boldsymbol{x}_t)$.

| METHOD | MASKING DISTRIBUTION $q(\boldsymbol{x}_t|\boldsymbol{x}_0)$ | WEIGHTING FUNCTION $w(t)$ | CONDITIONAL DISTRIBUTION $p_{\boldsymbol{\theta}}(\boldsymbol{x}_0^i \mid \boldsymbol{x}_t)$ |
|---|---|---|---|
| MASKGIT | UNIFORMLY MASK $\lceil N\gamma_t \rceil$ TOKENS W/O REPLACEMENT | $w(t) = 1$ | CATEGORICAL DISTRIBUTION |
| MAR | UNIFORMLY MASK $\lceil N\gamma_t \rceil$ TOKENS W/O REPLACEMENT | $w(t) = 1$ | DIFFUSION MODEL |
| MDM | MASK $N$ TOKENS INDEPENDENTLY WITH RATIO $\gamma_t$ | $w(t) = \frac{\gamma_t'}{\gamma_t}$ | CATEGORICAL DISTRIBUTION |

framework, representing the noisy or partially masked image at time $t$. That is, the masking distribution $q_{\gamma_r}(\boldsymbol{x}_{\overline{\mathbf{M}}}|\boldsymbol{x})$ can be mapped to a specific instance of $q(\boldsymbol{x}_t|\boldsymbol{x}_0)$, characterized by the chosen mask scheduling function $\gamma_t$. *See the equivalence between these two masking distributions in Appendix A.* After aligning these two masking distributions, MaskGIT, MAR, and MDM can be expressed within a unified loss function, defined as:

$$\mathcal{L}(\boldsymbol{x}_0) = \int_{t_{\min}}^{t_{\max}} w(t) \mathbb{E}_{q(\boldsymbol{x}_t|\boldsymbol{x}_0)} \left[ \sum_{\{i|\boldsymbol{x}_t^i=[\text{M}]\}} - \log p_{\boldsymbol{\theta}}\left(\boldsymbol{x}_0^i \mid \boldsymbol{x}_t\right) \right] dt. \tag{7}$$

In this unified formulation, the key differences between the models primarily lie in the three components outlined in Table 1. We explain these components as follows:

**Masking distribution $q(\boldsymbol{x}_t|\boldsymbol{x}_0)$.** For MaskGIT and MAR, $\lceil N\gamma_t \rceil$ tokens are uniformly masked without replacement as [M]. For MDM, each of the $N$ tokens is masked with probability $\gamma_t$ independently.

**Weighting function $w(t)$.** The weight function $w(t)$ determines the importance of the loss at each time step. For MaskGIT and MAR, $w(t) = 1$; for MDM, $w(t) = \frac{\gamma_t'}{\gamma_t}$.

**Conditional distribution $p_{\boldsymbol{\theta}}\left(\boldsymbol{x}_0^i \mid \boldsymbol{x}_t\right)$.** For MaskGIT and MDM, the conditional distribution $p_{\boldsymbol{\theta}}\left(\boldsymbol{x}_0^i \mid \boldsymbol{x}_t\right)$ is modeled as a categorical distribution. In contrast, for MAR, we employ a diffusion model assisted by a latent variable $\boldsymbol{z}$, leading to the following formulation:

$$p_{\boldsymbol{\theta}}(x_0^i|\boldsymbol{x}_t) = \int \delta_{\boldsymbol{\theta}_1}(\boldsymbol{z}^i|\boldsymbol{x}_t) p_{\boldsymbol{\theta}_2}^{\text{diff}}(x_0^i|\boldsymbol{z}^i) d\boldsymbol{z}^i. \tag{8}$$

Here, $\delta_{\boldsymbol{\theta}_1}(\boldsymbol{z}^i|\boldsymbol{x}_t)$ represents the output of the mask prediction model with input $\boldsymbol{x}_t$, and $p_{\boldsymbol{\theta}_2}^{\text{diff}}(x_0^i|\boldsymbol{z}^i)$ donated the output of diffusion model conditioned on $\boldsymbol{z}^i$.

## 4. Investigating the Design Space of Training

Building upon the unified framework, we now explore various design choices within this formulation. Given the equivalence of masking distributions, we adopt MDM's as the default setting. Furthermore, to mitigate the information loss introduced by the discrete tokenizer (Van Den Oord

et al., 2017; Esser et al., 2021), we use a diffusion model to model the conditional distribution $p_{\boldsymbol{\theta}}(x_0^i|\boldsymbol{x}_t)$. Our exploration begins with the standard MDM, which utilizes a single encoder transformer architecture and a linear mask schedule, in addition to using the diffusion model to model the conditional distribution $p_{\boldsymbol{\theta}}\left(\boldsymbol{x}_0^i \mid \boldsymbol{x}_t\right)$.

**Mask schedule.** The first critical aspect of our exploration is the choice of $\gamma_t$, which determines the probability of masking each token during the forward process (See Appendix B for details). In this section, we use the weighting function of $w(t) = \frac{\gamma_t'}{\gamma_t}$, which is mainly used in MDM. We consider three mask schedules: (1) *Linear*: $\gamma_t = t$; (2) *Cosine*: $\gamma_t = \cos\left(\frac{\pi}{2}(1-t)\right)$; (3) *Exp*: $\gamma_t = 1 - \exp(-5t)$. The first two mask schedules are also mentioned in Shi et al. (2024), while the last one is our design to achieve a higher masking ratio during training. As shown in Fig. 2(a), the cosine schedule outperforms the linear schedule. We hypothesize that, due to the high information redundancy in images, the cosine schedule achieves a higher mask ratio during training, providing stronger learning signals and leading to improved performance. The exp schedule further increases the mask ratio but destabilizes MDM training, likely due to the persistently large weighting function $w(t)$, even at high mask ratios (see Fig. 5 for visualization of $w(t)$ and $\gamma_t$).

**Weighting function.** We consider two choices for $w(t)$. (1) $w(t) = \frac{\gamma_t'}{\gamma_t}$, as used in MDM; (2) $w(t) = 1$, as used in MaskGIT. Notably, the weighting function significantly affects the choice of mask schedule. For instance, using $w(t) = \frac{\gamma_t'}{\gamma_t}$ led to unstable training, particularly with the exp schedule. In contrast, as shown in Fig. 2(b), setting $w(t) = 1$ stabilized the training process and improved performance, similar to the phenomenon observed in DDPM (Ho et al., 2020); under this setting, the exp schedule yielded the best results. Therefore, we adopted this combination ($w(t) = 1$ and the exp schedule) as our default.

**Model Architecture.** We consider two model architectures: (1) A single-encoder transformer; (2) The MAE (He et al., 2022a) architecture, which decomposes the transformer into an encoder-decoder structure, where the encoder processes

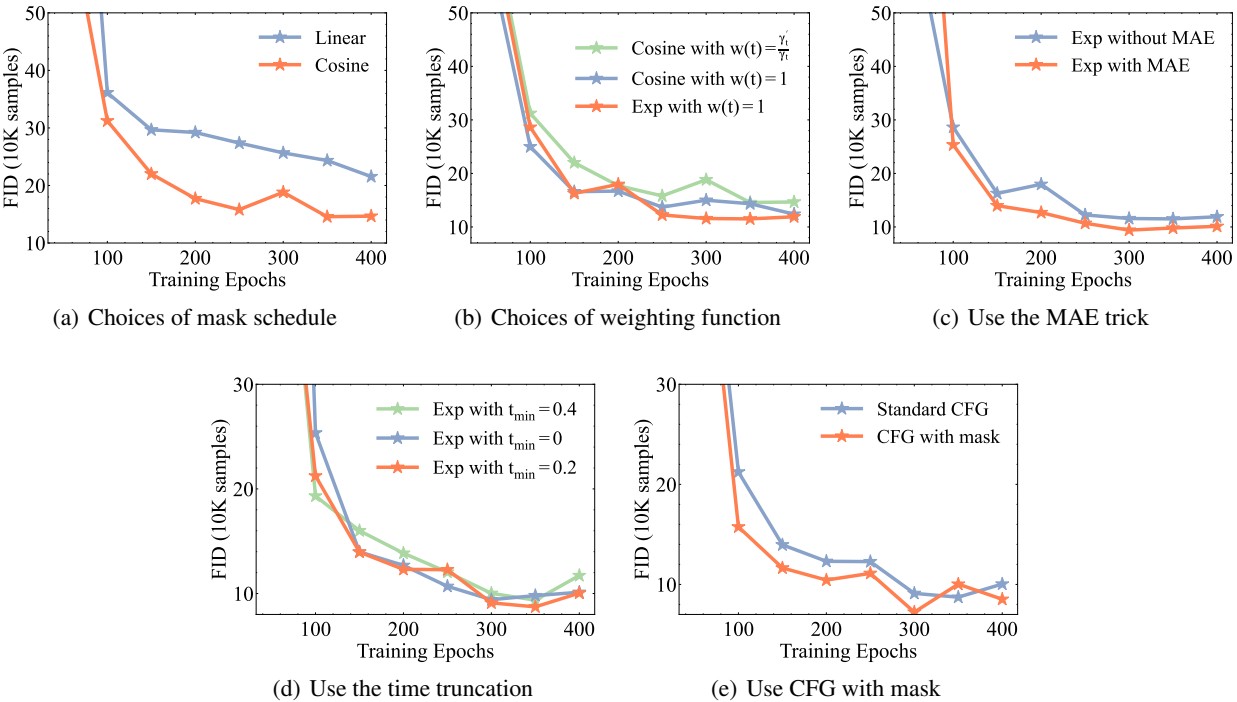

*Figure 2.* **Exploring the design space of training.** Orange solid lines indicate the preferred choices in each subfigure.

only unmasked tokens. The primary difference between these architectures is whether the encoder receives masked tokens as input. As shown in Fig. 2(c), under the exp schedule, the MAE architecture outperforms the single-encoder transformer. Interestingly, despite being originally designed for self-supervised learning, MAE retains its advantages in image generation. Therefore, unless otherwise specified, we adopt the MAE architecture as the default setting.

**Time Truncation.** To achieve a higher mask ratio during training, in addition to selecting a more concave function for $\gamma_t$, we can also use time truncation, which restricts the minimum value of $t$ to $t_{\min}$. We consider three choices: (1) $t_{\min} = 0$, the original design; (2) $t_{\min} = 0.2$; (3) $t_{\min} = 0.4$. As shown in Fig. 2(d), we observed that an appropriate time truncation ($t_{\min} = 0.2$) can be beneficial and accelerates training convergence. However, excessive truncation ($t_{\min} = 0.4$, where over 80% of image tokens are masked during training) provides no benefit and may even degrade performance compared to no time truncation. Unless otherwise noted, we adopt $t_{\min} = 0.2$ as the default setting.

**CFG with Mask.** Classifier-Free Guidance (CFG) (Ho & Salimans, 2022) is widely used for guiding continuous diffusion models and masked image generation. It combines outputs of a conditional model (with class information) and an unconditional model (without class information) to improve alignment with the conditional output. In standard CFG, the unconditional model typically receives a learn-

able fake class token as input. Unsupervised classifier-free guidance was initially developed for text generation (Nie et al., 2024), a process involving the unconditional model receiving a special mask token as input. Inspired by this method, our paper adapts it for image generation. We term this adapted approach *CFG with Mask* to emphasize its focus on masked image generation. As shown in Fig. 2(e), CFG with mask improves generation performance compared to standard CFG. Notably, here we use only simple conditional generation without guidance, our results suggest that using a fake class token negatively impacts the conditional generation performance of MDM. Thus, we adopt CFG with mask as the default setting.

## 5. Investigating the Design Space of Sampling

In the previous section, we carefully explore the training design space. In the following sections, we investigate the sampling design space. On one hand, we expect the model's performance to improve as the number of mask prediction steps increases. On the other hand, we aim to maintain strong performance even with a low number of mask prediction steps (e.g., 16).

### 5.1. Mask Schedule during Sampling

During training, we observe that the exp schedule achieves the best performance. However, during sampling, different

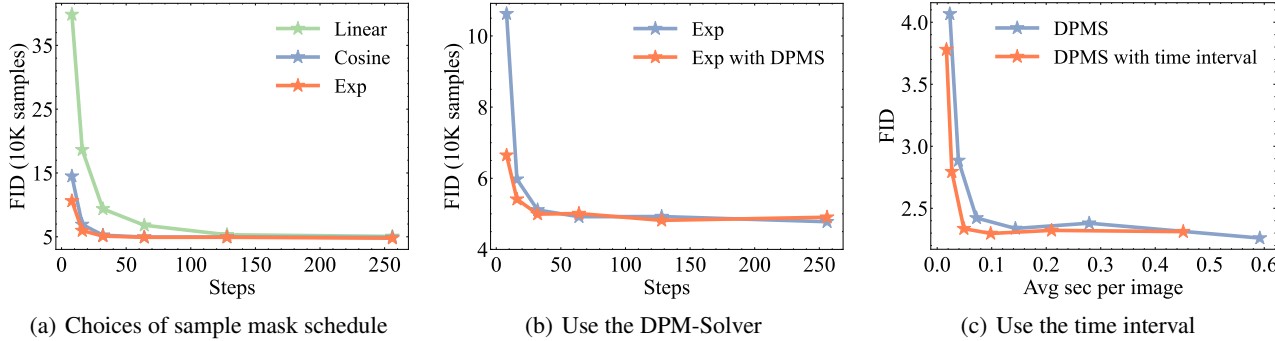

(a) Choices of sample mask schedule  (b) Use the DPM-Solver  (c) Use the time interval

*Figure 3.* **Exploring the design space of sampling.** For each plot, points from left to right correspond to an increasing number of mask prediction steps: 8, 16, 32, and up to 256. In each subfigure, DPM-Solver is donated as DPMS. (a) The exp schedule outperforms others by predicting fewer tokens early. (b) DPM-Solver performs better with fewer prediction steps. (c) The time interval maintains performance while reducing sampling cost for each mask prediction step, particularly for high mask prediction steps.

schedules may be employed. We are interested in identifying which mask schedule can achieve both of our goals.

To this end, we first conduct a simulation experiment (see details in Appendix B.2) to compare the number of tokens predicted during each mask prediction step across different mask schedules. We observe that the linear schedule predicts a nearly constant number of tokens per step, while the cosine schedule predicts fewer tokens early in the process and progressively more later. This observation aligns with the findings reported in Shi et al. (2024). Besides, the exp schedule predicts even fewer tokens initially, with a more gradual increase as the process continues. As shown in Fig. 3(a), we observe that each mask schedule benefits more prediction steps. Moreover, for low mask prediction steps (e.g., 8 or 16), the exp schedule outperforms the cosine schedule, which in turn outperforms the linear schedule. This suggests that, in the early stages of sampling, predicting fewer tokens may contribute to improved performance at lower mask prediction steps. Thus, we adopt the exp schedule as our default for sampling unless otherwise specified.

### 5.2. The Sampling Method of Diffusion Loss

We use the diffusion loss to model the distribution of $p_{\boldsymbol{\theta}}\left(\boldsymbol{x}_0^i \mid \boldsymbol{x}_t\right)$. Previously, we follow MAR (Li et al., 2024) and use DDPM (Ho et al., 2020) sampling method with 100 diffusion steps. Additionally, MAR employs the temperature $\tau$ sampling method from ADM (Dhariwal & Nichol, 2021) to scale the noise by $\tau$, which requires careful tuning for optimal performance.

In contrast, DPM-Solver (Lu et al., 2022a;b) is a training-free, fast ODE sampler that accelerates the diffusion sampling process and converges faster with fewer steps. Interestingly, although DPM-Solver is designed for accelerating the diffusion process, we observe that, with low mask prediction steps, it outperforms DDPM, as shown in Fig. 3(b).

For example, with 8 mask prediction steps, DPM-Solver achieves an FID of 6.6, while DDPM, with a temperature of 1.0, achieves an FID of 10.6. We hypothesize that for low mask prediction steps, DDPM requires careful temperature tuning, whereas DPM-Solver, being an ODE sampler, does not require such adjustments. Moreover, DPM-Solver achieves good performance with fewer than 15 diffusion steps, while DDPM requires 100 diffusion steps. Therefore, unless specified, we default to DPM-Solver.

### 5.3. Time Interval for Classifier Free Guidance

Previously, we adopt a linear CFG schedule following MAR (Li et al., 2024), where the CFG value gradually increased from 0 to the target value during the mask prediction process. With a constant CFG schedule, we find that the generation performance is highly sensitive to the CFG value, as shown in Fig. 7. We hypothesize that, for MDM, token generation is irreversible—once a token is generated, it cannot be modified. Therefore, a strong guide in the early stages may reduce the variation in the results, leading to a higher FID. This is similar to our earlier observation with the linear mask schedule, where generating too many incorrect tokens early can cause error accumulation and degrade the performance. We conduct an experiment with a total of 256 sample tokens and 16 mask prediction steps (see details in Appendix C) to validate our hypothesis. Let $s_i$ and $t_i$ denote the endpoint and start of the $i$-th step in the mask prediction process. We apply CFG if $s_i \in [\text{cfg}_{t_{\min}}, \text{cfg}_{t_{\max}}]$; otherwise, we use simple conditional generation. As shown in Fig. 8(a), when $\text{cfg}_{t_{\min}} < \text{cfg}_{t_{\max}} \leq 0.5$, we achieve a relatively low FID, supporting our hypothesis. In particular, the best performance is achieved when $\text{cfg}_{t_{\min}} = 0.1$ and $\text{cfg}_{t_{\max}} = 0.3$, using only 60% of the NFE (the number of function evaluations) compared to standard CFG. Specifically, for standard CFG, NFE = $16 \times 2$, while for the time interval, NFE $\approx 16 + 16 \times (0.3 - 0.1)$.

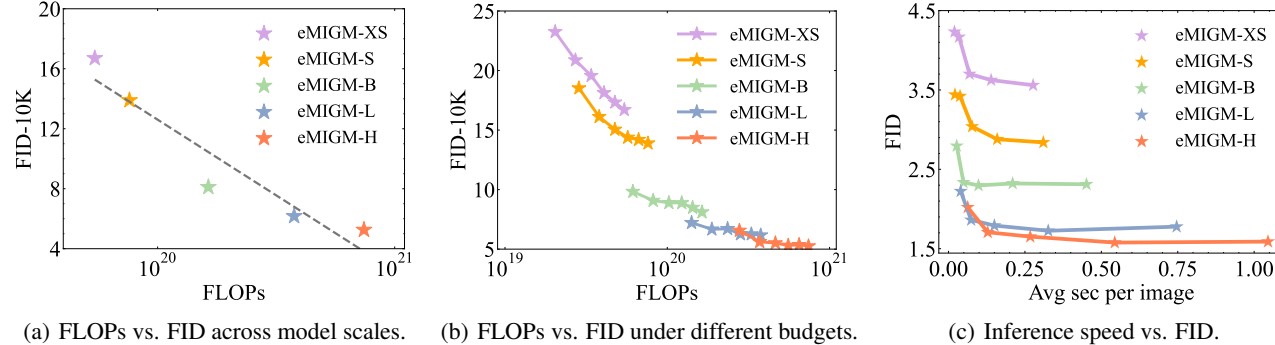

(a) FLOPs vs. FID across model scales.  (b) FLOPs vs. FID under different budgets.  (c) Inference speed vs. FID.

*Figure 4.* **Scalability of eMIGM.** (a) A negative correlation demonstrates that eMIGM benefits from scaling. (b) Larger models are more training-efficient (i.e., achieving better sample quality with the same training FLOPs). (c) Larger models are more sampling-efficient (i.e., achieving better sample quality with the same inference time).

*Table 2.* **Image generation results on ImageNet** $256 \times 256$. † denotes results taken from MaskGIT (Chang et al., 2022), and ⋆ indicates results that require assistance from the self-supervised model. *With* $42\%$ *of function evaluations (NFE), eMIGM-H achieves performance comparable to the state-of-the-art diffusion model REPA (Yu et al., 2024).* We **bold** the best result under each method and underline the second-best result.

| METHOD | NFE ($\downarrow$) | FID ($\downarrow$) | #Params |
|---|---|---|---|
| **Diffusion models** | | | |
| ADM-G (Dhariwal & Nichol, 2021) | 250× 2 | 4.59 | 554M |
| ADM-G-U (Dhariwal & Nichol, 2021) | 750 | 3.94 | 554M |
| LDM-4-G (Rombach et al., 2022) | 250× 2 | 3.60 | 400M |
| VDM++ (Kingma & Gao, 2024) | 512×2 | 2.40 | 2B |
| SimDiff (Hoogeboom et al., 2023) | 512×2 | 2.44 | 2B |
| U-ViT-H/2 (Bao et al., 2023) | 50×2 | 2.29 | 501M |
| DiT-XL/2 (Peebles & Xie, 2023) | 250×2 | 2.27 | 675M |
| Large-DiT (Alpha-VLLM, 2024) | 250×2 | 2.10 | 3B |
| Large-DiT (Alpha-VLLM, 2024) | 250×2 | 2.28 | 7B |
| SiT-XL (Ma et al., 2024) | 250×2 | 2.06 | 675M |
| D$_{\text{IFFU}}$SSM-XL-G (Yan et al., 2024) | 250×2 | 2.28 | 660M |
| DiffiT (Hatamizadeh et al., 2025) | 250×2 | 1.73 | 561M |
| REPA (Yu et al., 2024)⋆ | 250×1.7 | **1.42** | 675M |
| **ARs** | | | |
| VQGAN (Esser et al., 2021)† | 256 | 18.65 | 227M |
| VAR-$d16$ (Tian et al., 2024) | 10×2 | 3.30 | 310M |
| VAR-$d20$ (Tian et al., 2024) | 10×2 | 2.57 | 600M |
| VAR-$d24$ (Tian et al., 2024) | 10×2 | 2.09 | 1B |
| VAR-$d30$ (Tian et al., 2024) | 10×2 | **1.92** | 2B |

| METHOD | NFE ($\downarrow$) | FID ($\downarrow$) | #Params |
|---|---|---|---|
| **GANs** | | | |
| BigGAN (Brock, 2018) | 1 | 6.95 | - |
| StyleGAN-XL (Sauer et al., 2022) | 1×2 | 2.30 | - |
| **Masked models** | | | |
| MaskGIT (Chang et al., 2022)† | 8 | 6.18 | 227M |
| MAR-B (Li et al., 2024) | 256×2 | 2.31 | 208M |
| MAR-L (Li et al., 2024) | 256×2 | 1.78 | 479M |
| MAR-H (Li et al., 2024) | 256×2 | **1.55** | 943M |
| **Ours** | | | |
| eMIGM-XS | 16×1.2 | 4.23 | 69M |
| eMIGM-S | 16×1.2 | 3.44 | 97M |
| eMIGM-B | 16×1.2 | 2.79 | 208M |
| eMIGM-L | 16×1.2 | 2.22 | 478M |
| eMIGM-H | 16×1.2 | 2.02 | 942M |
| eMIGM-XS | 128×1.4 | 3.62 | 69M |
| eMIGM-S | 128×1.4 | 2.87 | 97M |
| eMIGM-B | 128×1.35 | 2.32 | 208M |
| eMIGM-L | 128×1.4 | 1.72 | 478M |
| eMIGM-H | 128×1.4 | 1.57 | 942M |

As shown in Fig. 3(c), we observe that the time interval maintains performance at each mask prediction step while reducing sampling time. This demonstrates its efficiency and effectiveness. Therefore, we adopt the time interval for all subsequent experiments in this paper.

## 6. Experiments

By fully considering the design space mentioned above, we evaluate eMIGM on ImageNet $256 \times 256$ and ImageNet $512 \times 512$ (Deng et al., 2009), benchmarking the sample quality using Fréchet Inception Distance (FID) (Heusel et al., 2017). See experiment settings in Appendix D.

### 6.1. Larger Models Are Training and Sampling Efficient

First, to demonstrate the scaling properties of eMIGM, we plot the FID-10K at 400 training epochs for different model sizes of eMIGM against training FLOPs. As shown in Fig. 4(a), we observe a negative correlation between training FLOPs and FID-10K, indicating that eMIGM benefits from scaling. Second, for different model sizes of eMIGM, we scale the FLOPs and analyze the FID-10K in relation to training FLOPs. As shown in Fig. 4(b), for each model size of eMIGM, as training epochs and training FLOPs increase, performance also improves. Additionally, we observe that for the same training FLOPs, larger eMIGM models achieve better performance. For instance, eMIGM-L outperforms

*Table 3.* **Image generation results on ImageNet** $512 \times 512$. [†] denotes results taken from MaskGIT (Chang et al., 2022). [‡] denotes results obtained using Guidance Interval (Kynkäänniemi et al., 2024). *With 20 function evaluations (NFE), eMIGM-L outperforms strong visual autoregressive models VAR (Tian et al., 2024). When the NFE increases to 80, eMIGM-L surpasses the strong diffusion model EDM2 (Karras et al., 2024).* We **bold** the best result under each method and underline the second-best result.

| METHOD | NFE ($\downarrow$) | FID ($\downarrow$) | #Params |
|---|---|---|---|
| **Diffusion models** | | | |
| ADM-G (Dhariwal & Nichol, 2021) | $250\times 2$ | 7.72 | 559M |
| ADM-G-U (Dhariwal & Nichol, 2021) | 750 | 3.85 | 559M |
| VDM++ (Kingma & Gao, 2024) | $512\times 2$ | 2.65 | 2B |
| SimDiff (Hoogeboom et al., 2023) | $512\times 2$ | 3.02 | 2B |
| U-ViT-H/4 (Bao et al., 2023) | $50\times 2$ | 4.05 | 501M |
| DiT-XL/2 (Peebles & Xie, 2023) | $250\times 2$ | 3.04 | 675M |
| Large-DiT (Alpha-VLLM, 2024) | $250\times 2$ | 2.52 | 3B |
| SiT-XL (Ma et al., 2024) | $250\times 2$ | 2.62 | 675M |
| EDM2-XXL (Karras et al., 2024) | $32\times 2$ | 1.81 | 1.5B |
| EDM2-XXL (Kynkäänniemi et al., 2024)[‡] | $32\times 1.2$ | **1.40** | 1.5B |
| **Consistency models** | | | |
| sCT-XXL (Lu & Song, 2024) | 2 | 3.76 | 1.5B |
| sCD-XXL (Lu & Song, 2024) | 2 | 1.88 | 1.5B |
| **GANs** | | | |
| BigGAN (Brock, 2018) | 1 | 8.43 | - |
| StyleGAN-XL (Sauer et al., 2022) | $1\times 2$ | 2.41 | - |

| METHOD | NFE ($\downarrow$) | FID ($\downarrow$) | #Params |
|---|---|---|---|
| **ARs** | | | |
| VQGAN (Esser et al., 2021)[†] | 1024 | 26.52 | 227M |
| VAR-$d36$-s (Tian et al., 2024) | $10\times 2$ | 2.63 | 2.3B |
| **Masked models** | | | |
| MaskGIT (Chang et al., 2022)[†] | 12 | 7.32 | 227M |
| MAR (Li et al., 2024) | $256\times 2$ | **1.73** | 481M |
| **Ours** | | | |
| eMIGM-XS | $16\times 1.2$ | 4.63 | 104M |
| eMIGM-S | $16\times 1.2$ | 3.65 | 132M |
| eMIGM-B | $16\times 1.2$ | 2.78 | 244M |
| eMIGM-L | $16\times 1.2$ | 2.19 | 478M |
| eMIGM-XS | $64\times 1.25$ | 4.45 | 104M |
| eMIGM-S | $64\times 1.25$ | 3.29 | 132M |
| eMIGM-B | $64\times 1.25$ | 2.31 | 244M |
| eMIGM-L | $64\times 1.25$ | 1.77 | 478M |

eMIGM-B with approximately $10^{20}$ FLOPs. Third, we observed the inference-time scaling behavior of eMIGM. As shown in Fig. 4(c), we plot the performance of different eMIGM model sizes across various mask prediction steps (ranging from 16 to 256). The speed is measured using a single A100 GPU with a batch size of 256. We observe that as the number of prediction steps increases, each model size of eMIGM achieves better performance, particularly for smaller models (i.e., eMIGM-XS and eMIGM-S). For larger model sizes, a similar best performance is reached with just 64 steps. Additionally, we also find that larger eMIGM models achieve better performance while maintaining similar inference speeds. For example, at a speed of about 0.2 seconds per image, eMIGM-L achieves a strong FID of 1.8, outperforming eMIGM-B with an FID of 2.3.

### 6.2. Image Generation on ImageNet

In Tab. 2, we compare eMIGM with state-of-the-art generative models on **ImageNet** $256 \times 256$. Notably, in Tab. 2 and Tab. 3, we list only the NFE of eMIGM's transformer component. When measured on a single A100 GPU with a batch size of 256, we found that the MLP diffusion block introduces approximately 14% additional computational overhead beyond the NFE of the main transformer. However, since the transformer component remains the primary computational bottleneck, NFE continues to be a valid efficiency metric. By exploring the design space of sampling, eMIGM with few NFEs (approximately 20) outperforms VAR (Tian et al., 2024) with a similar model size. Specifically, eMIGM-B achieves an FID of 2.79 with only 208M parameters, while VAR-d16 achieves an FID of 3.30 with

310M parameters. Notably, as we increase the NFE, all of our models consistently show significant improvements in generation performance. For instance, eMIGM-L achieves an FID of 1.72 with 180 NFEs, compared to an FID of 2.22 with 20 NFEs. By increasing the NFE, eMIGM-L, despite having only 478M parameters, outperforms the best VAR-d30, which achieves an FID of 1.92 with 2B parameters. Lastly, our more powerful eMIGM-H achieves an FID of 1.57 with just 180 NFEs, outperforming strong diffusion models such as Large-DiT (Alpha-VLLM, 2024) and DiffiT (Hatamizadeh et al., 2025). eMIGM-H is also comparable to the best diffusion models REPA (Yu et al., 2024), which require 425 sequential steps and the assistance of the self-supervised model. Furthermore, compared to the state-of-the-art GAN model StyleGAN-XL (Sauer et al., 2022), eMIGM-B achieves superior performance. We also present more evaluation metrics on Tab. 8 in the appendix.

We also evaluate eMIGM on higher resolution images (i.e., $512 \times 512$) in Tab. 3. Specifically, with similar NFEs, eMIGM-L (with only 478M parameters) achieves an FID of 2.19, outperforming the strong generative model VAR (Tian et al., 2024) (with 2.3B parameters), which achieves an FID of 2.63. Furthermore, compared to the strong diffusion model EDM2 (Karras et al., 2024), eMIGM-L achieves an FID of 1.77, outperforming EDM2's FID of 1.81. These quantitative results demonstrate that eMIGM achieves excellent generation performance and high sampling efficiency across diverse resolutions. However, when using the guidance interval (Kynkäänniemi et al., 2024), EDM2-XXL achieves superior performance while needing more parameters. A comparison of the sampling speeds for eMIGM and EDM2 (Karras et al., 2024) is also presented in Tab. 10.

Furthermore, when compared to MAR, eMIGM-L achieves competitive performance while using an NFE of less than 20%.

## 7. Related Work

**Visual generation.** Modern visual generation models primarily fall into four categories: GANs (Goodfellow et al., 2014; Brock, 2018; Sauer et al., 2022), diffusion models (Song et al., 2020; Sohl-Dickstein et al., 2015; Ho et al., 2020), masked prediction models (Chang et al., 2022; Li et al., 2023; 2024; Bai et al., 2024; Shao et al., 2024; Ni et al., 2024), and autoregressive models (Esser et al., 2021; Tian et al., 2024; Sun et al., 2024; Tang et al., 2024). The most related works to our study are MaskGIT (Chang et al., 2022) and MAR (Li et al., 2024). We provide a unified framework that integrates both approaches and systematically explore the impact of each component. Additionally, guidance interval (Kynkäänniemi et al., 2024) and CADS (Sadat et al., 2023) also observed that strong guidance early in the process negatively affects diversity. Therefore, they proposed sampling strategies to adjust the guidance application during sampling. Besides, Wang et al. (2024) also analyses the schedule of classifier-free guidance in continuous diffusion models. However, unlike our proposed time interval, which applies guidance at the token level, their methods operate at different noise levels of the entire image. Besides, our proposed time interval is motivated by MDM's unique irreversible token generation constraint. Furthermore, Shao et al. (2024) proposed an enhanced inference technique to improve the speed and performance of masked image generative models such as MaskGIT (Chang et al., 2022) and Meissonic (Bai et al., 2024). Their technique is orthogonal to our method and can also be applied to our work.

**Masked discrete diffusion models.** Recently, masked discrete diffusion models (Austin et al., 2021; Campbell et al., 2022), a special case of discrete diffusion models (Sohl-Dickstein et al., 2015; Hoogeboom et al., 2021), have achieved remarkable progress in various domains, including text generation (He et al., 2022b; Lou et al., 2024a; Shi et al., 2024; Sahoo et al., 2024; Ou et al., 2024; Zheng et al., 2023; Chen et al., 2023; Gat et al., 2024; Nie et al., 2024), music generation (Sun et al., 2023), protein design (Campbell et al., 2024), and image generation (Hu & Ommer, 2024).

## 8. Conclusion

In this paper, we present a single framework to unify masked image generation models and masked diffusion models and carefully examine each component of design space to achieve efficient and high-quality image generation. Empirically, we demonstrate that eMIGM can achieve comparable performance with the state-of-the-art continuous diffusion models with fewer NFEs. We believe that eMIGM will inspire future research in masked image generation.

## Acknowledgements

This work was sponsored by the Beijing Nova Program (No. 20230484416); National Natural Science Foundation of China (No. 92470118); Beijing Natural Science Foundation (No. L247030); the ant Group Research Fund.

## Impact Statement

We introduce eMIGM, a powerful generative model that significantly accelerates the sampling speed while maintaining high image quality. However, this increased efficiency may increase the potential for misuse of generated images. To mitigate this, watermarks can be embedded into the generated images without affecting the generation quality, helping to prevent misuse and verify if an image is generated.

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

*Table 4.* **Mask schedule formulations.**

| Mask schedule | $\gamma_t$ | $\frac{-\gamma'_t}{\gamma_t}$ |
|---|---|---|
| Linear | $t$ | $-\frac{1}{t}$ |
| Cosine | $\cos\left(\frac{\pi}{2}(1-t)\right)$ | $-\frac{\pi}{2}\tan\left(\frac{\pi}{2}(1-t)\right)$ |
| Exp | $1-\exp(-5t)$ | $-\frac{5\exp(-5t)}{1-\exp(-5t)}$ |

## A. Equivalence of the masking strategies of MaskGIT and MDM

In this section, we demonstrate that the masking strategies of MaskGIT and MDM are equivalent in expectation. MaskGIT first samples a ratio $r$ from $[0,1]$ and then uniformly masks $\lceil N\gamma_r \rceil$ tokens of $\boldsymbol{x}$ as [M]. In contrast, for MDM, each token is independently masked as [M] with probability $\gamma_t$.

First, for MDM, the cross-entropy loss in Equation (4) has multiple equivalent forms (Ou et al., 2024). To facilitate better understanding, we reformulate Equation (4) as an expectation over $t$:

$$\mathcal{L}(\boldsymbol{x_0}) = \mathbb{E}_{t\sim U[0,1]}\mathbb{E}_{q(\boldsymbol{x}_t|\boldsymbol{x}_0)}\left[\frac{\gamma'_t}{\gamma_t}\sum_{\{i|\boldsymbol{x}_t^i=[\text{M}]\}} -\log p_{\boldsymbol{\theta}}(\boldsymbol{x}_0^i|\boldsymbol{x}_t)\right]. \tag{9}$$

As an example, we consider the linear mask schedule, where $\gamma_t = t$. In this formulation, the forward process involves independently masking each token based on a uniformly sampled $t$. Under this setting, the loss simplifies to:

$$\mathcal{L}(\boldsymbol{x_0}) = \mathbb{E}_{t\sim U[0,1]}\mathbb{E}_{q(\boldsymbol{x}_t|\boldsymbol{x}_0)}\left[\frac{1}{t}\sum_{\{i|\boldsymbol{x}_t^i=[\text{M}]\}} -\log p_{\boldsymbol{\theta}}(\boldsymbol{x}_0^i|\boldsymbol{x}_t)\right]. \tag{10}$$

For MaskGIT, the number of masked tokens $l$ is sampled from a uniform distribution $U[1,N]$, after which $l$ tokens in $\boldsymbol{x}_0$ are randomly masked as [M]. Under this scheme, the loss function can be rewritten as:

$$\mathcal{L}(\boldsymbol{x_0}) = \mathbb{E}_{l\sim U[1,N]}\mathbb{E}_{q(\boldsymbol{x}_l|\boldsymbol{x}_0)}\left[\frac{1}{\frac{l}{N}}\sum_{\{i|\boldsymbol{x}_l^i=[\text{M}]\}} -\log p_{\boldsymbol{\theta}}(\boldsymbol{x}_0^i|\boldsymbol{x}_l)\right]. \tag{11}$$

As shown in Ou et al. (2024), Equation (11) and Equation (10) are equivalent in expectation. In this paper, we adopt the formulation of Equation (4) with an exponential mask schedule as the default setting.

## B. Mask schedules

### B.1. Formulations and Illustrations of Mask Schedules

We present different choices of mask schedules in Fig. 5 and Tab. 4. The linear schedule achieves the best empirical performance in text generation, as demonstrated in previous work (Lou et al., 2024b; Sahoo et al., 2024; Shi et al., 2024). In comparison to the linear schedule, the cosine and exp schedules mask more tokens during the forward process of MDM.

### B.2. Sampling Simulator Experiment

During sampling, we conducted a simulation experiment with a total of 256 sample tokens and 16 sampling steps. Therefore, the temporal interval $[0,1]$ is discretized into 16 equally sized segments for sampling purposes. Let $s_i$ and $t_i$ represent the endpoint and starting point of the $i$-th segment, respectively, where $i \in \{1, 2, \ldots, 16\}$. The indexing is defined such that $t_1$ corresponds to the start of the first segment. Specifically, the endpoints are defined as $s_i = \frac{16-i}{16}$ and the starting points as $t_i = \frac{16-i+1}{16}$. In each step $i$, the prediction for each token is made with a probability of $\frac{\gamma_{t_i}-\gamma_{s_i}}{\gamma_{s_i}}$, as given by Equation (6).

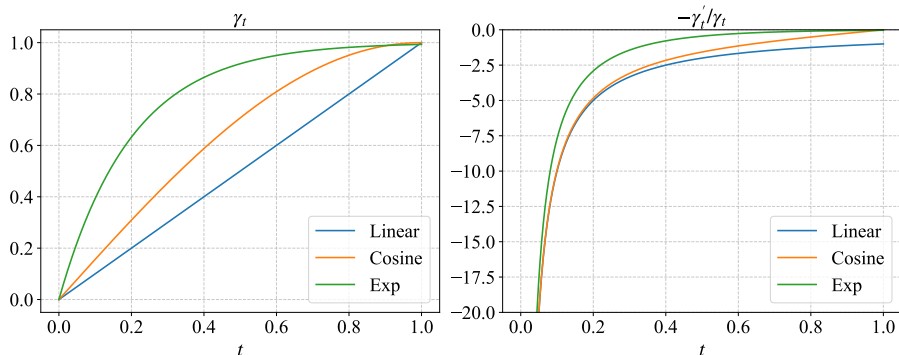

*Figure 5.* **Different choices of mask schedules.** Left: $\gamma_t$ (i.e., the probability that each token is masked during the forward process). Right: Weight of the loss in MDM.

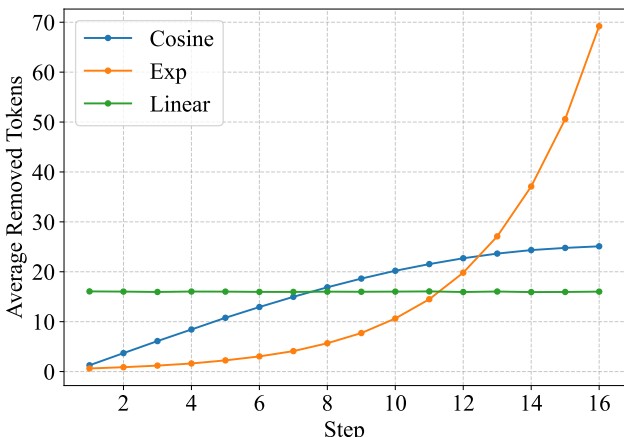

*Figure 6.* **Comparison of mask removal** for different sample mask schedule.

We simulated the process 10,000 times and calculated the average number of tokens predicted in each step. The experimental results are shown in Fig. 6.

We observed the following trends: For the linear schedule, the model predicts almost the same number of tokens in each step. In contrast, for the cosine schedule, the model predicts fewer tokens in the earlier steps and more tokens in the later steps. Compared to the cosine schedule, the exp schedule predicts even fewer tokens in the earlier steps and progressively more tokens in the later steps.

## C. Time Interval for Classifier Free Guidance

To validate our hypothesis that an excessively strong guide in the early stages may drastically reduce the variation in generated samples, leading to a higher FID, we conducted an experiment with a total of 256 sample tokens and 16 sampling steps. A more detailed description of the sampling procedure can be found in Appendix B.2. Let $s_i$ and $t_i$ represent the endpoint and starting point of the $i$-th sampling step, respectively. We define $t_{min}$ and $t_{max}$ for CFG. If $s_i \in [t_{min}, t_{max}]$, we apply CFG to guide the sampling; otherwise, we do not use CFG and rely solely on simple conditional generation. As shown in Fig. 8, we observe that when $t_{min} = 0$ and $t_{max} = 1$, the FID value is 22.48, demonstrating low variation in the generated samples. Additionally, in the top left corner of Fig. 8(a) (i.e., when $t_{min} < t_{max} \le 0.5$), we achieve a relatively low FID (indicating higher variation), which supports our hypothesis and encourages the application of CFG guidance only during the later stages of sampling.

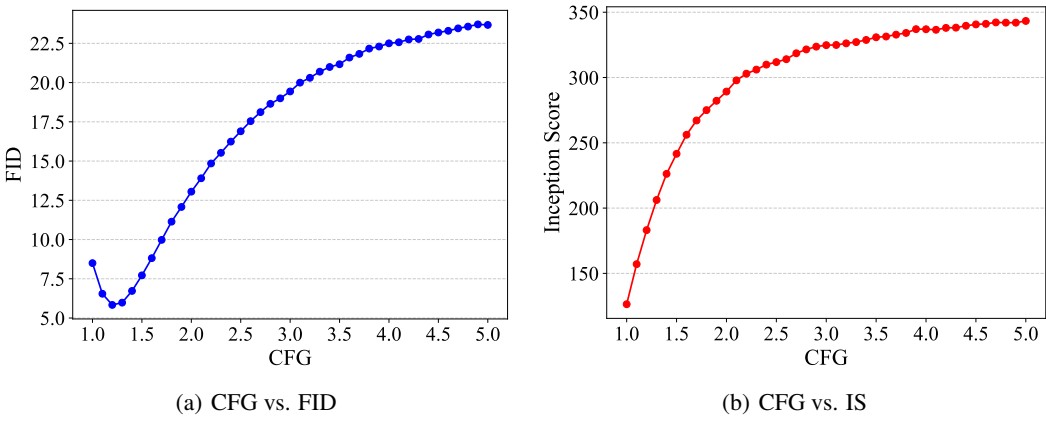

(a) CFG vs. FID

(b) CFG vs. IS

*Figure 7.* **Generation performance is sensitive to the CFG value** when using the constant schedule.

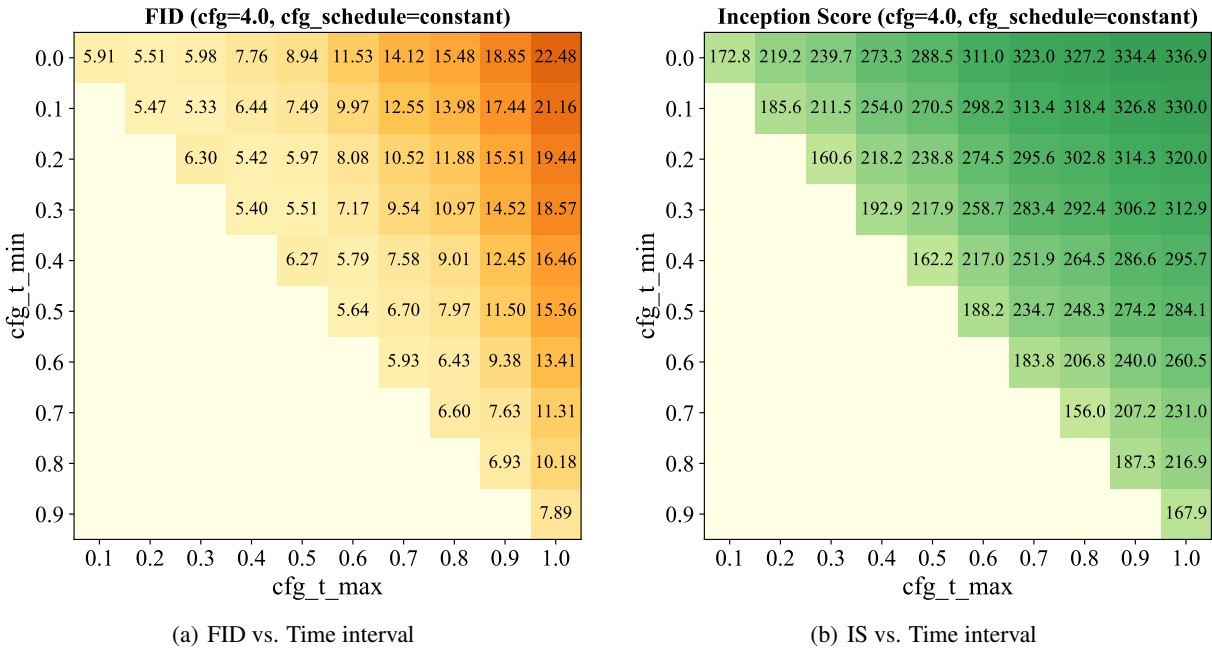

(a) FID vs. Time interval

(b) IS vs. Time interval

*Figure 8.* **Performance across different time intervals.** Subplots show (a) FID and (b) Inception Score(IS).

*Table 5.* **The code links and licenses.**

| Method | Link | License |
|---|---|---|
| MAR | https://github.com/LTH14/mar | MIT License |
| DPM-Solver | https://github.com/LuChengTHU/dpm-solver | MIT License |
| DC-AE | https://github.com/mit-han-lab/efficientvit | Apache-2.0 license |

# D. Experiment settings and results

We implement eMIGM upon the official code of MAR (Li et al., 2024), DC-AE (Chen et al., 2024), DPM-Solver (Lu et al., 2022a;b), whose code links and licenses are presented in Tab. 5.

**Image Tokenizer.** For ImageNet $256 \times 256$, we use the same KL-16 image tokenizer as in MAR (Li et al., 2024), which has a stride of 16. That is, for an image of size $256 \times 256$, it outputs an image token sequence of length $16 \times 16$, with each

Table 6. **Training configurations of models on ImageNet 256×256.**

| | Model Size | | | | |
|---|---|---|---|---|---|
| | XS | S | B | L | H |
| *Architecture Configurations* | | | | | |
| Transformer blocks | 20 | 24 | 24 | 32 | 40 |
| Transformer width | 448 | 512 | 768 | 1024 | 1280 |
| MLP blocks | 3 | 3 | 6 | 8 | 12 |
| MLP width | 1024 | 1024 | 1024 | 1280 | 1536 |
| Params (M) | 69 | 97 | 208 | 478 | 942 |
| *Training Hyperparameters* | | | | | |
| Epochs | 800 | 800 | 800 | 800 | 800 |
| Learning rate | 4.0e-4 | 4.0e-4 | 8.0e-4 | 8.0e-4 | 8.0e-4 |
| Batch size | 1024 | 1024 | 2048 | 2048 | 2048 |
| Adam $\beta_1$ | 0.9 | 0.9 | 0.9 | 0.9 | 0.9 |
| Adam $\beta_2$ | 0.95 | 0.95 | 0.95 | 0.95 | 0.95 |

Table 7. **Training configurations of models on ImageNet 512×512.**

| | Model Size | | | |
|---|---|---|---|---|
| | XS | S | B | L |
| *Architecture Configurations* | | | | |
| Transformer blocks | 20 | 24 | 24 | 32 |
| Transformer width | 448 | 512 | 768 | 1024 |
| MLP blocks | 6 | 6 | 8 | 8 |
| MLP width | 1280 | 1280 | 1280 | 1280 |
| Params (M) | 104 | 132 | 244 | 478 |
| *Training Hyperparameters* | | | | |
| Epochs | 800 | 800 | 800 | 800 |
| Learning rate | 4.0e-4 | 4.0e-4 | 8.0e-4 | 8.0e-4 |
| Batch size | 1024 | 1024 | 2048 | 2048 |
| Adam $\beta_1$ | 0.9 | 0.9 | 0.9 | 0.9 |
| Adam $\beta_2$ | 0.95 | 0.95 | 0.95 | 0.95 |

token having a dimensionality of 16. For ImageNet $512 \times 512$, we use the DC-AE-f32 tokenizer (Chen et al., 2024) for efficiency, which has a stride of 32, and each token has a dimensionality of 32.

**Classifier-Free Guidance (CFG).** In the original CFG, during training, the class condition is replaced with a fake class token with a probability of 10%. During sampling, the prediction model takes both the class token and the fake class token as input, generating outputs $z_c$ and $z_u$. Conceptually, CFG encourages the generated image to align more closely with the result conditioned on $z_c$ while deviating from the result conditioned on $z_u$. For CFG with Mask, we replace the fake class token with a masked token as the input for unconditional generation. We use a constant CFG schedule and the time interval strategy in our main results presented in Tab. 2 and Tab. 3, achieving excellent performance while significantly reducing the sampling cost. Moreover, we observed that with the time interval strategy, we can use a consistently high CFG value to guide generation at each prediction step, eliminating the need for CFG value sweeping.

**Training Settings.** The detailed training settings for ImageNet $256 \times 256$ and ImageNet $512 \times 512$ are provided in Tab. 6 and Tab. 7, respectively.

*Table 8.* **Image generation results on ImageNet** $256 \times 256$.

| Method | NFE | FID↓ | sFID↓ | IS↑ | Precision↑ | Recall↑ |
|---|---|---|---|---|---|---|
| VAR-d30 (Tian et al., 2024) | 10×2 | 1.92 | - | 323.1 | 0.82 | 0.59 |
| REPA (Yu et al., 2024) | 250×1.7 | 1.42 | 4.70 | 305.7 | 0.80 | 0.65 |
| eMIGM-XS | 16×1.2 | 4.23 | 5.74 | 218.63 | 0.79 | 0.50 |
| eMIGM-S | 16×1.2 | 3.44 | 5.31 | 244.16 | 0.80 | 0.53 |
| eMIGM-B | 16×1.2 | 2.79 | 5.20 | 284.62 | 0.82 | 0.54 |
| eMIGM-L | 16×1.2 | 2.22 | 4.80 | 291.62 | 0.80 | 0.59 |
| eMIGM-H | 16×1.2 | 2.02 | 4.66 | 299.36 | 0.80 | 0.60 |
| eMIGM-XS | 128×1.4 | 3.62 | 5.47 | 224.91 | 0.80 | 0.51 |
| eMIGM-S | 128×1.4 | 2.87 | 5.53 | 254.48 | 0.80 | 0.54 |
| eMIGM-B | 128×1.35 | 2.32 | 4.63 | 278.97 | 0.81 | 0.57 |
| eMIGM-L | 128×1.4 | 1.72 | 4.63 | 304.16 | 0.80 | 0.60 |
| eMIGM-H | 128×1.4 | 1.57 | 4.68 | 305.99 | 0.80 | 0.63 |

*Table 9.* **Ablation study** on different mask schedules, reporting FID scores.

| Epoch | Linear | Cosine | Exp | Log-Exp |
|---|---|---|---|---|
| 100 | 38.66 | 24.99 | 28.63 | 25.38 |
| 200 | 30.55 | 16.70 | 17.97 | 11.81 |
| 300 | 24.55 | 15.00 | 11.57 | 12.48 |
| 400 | 24.96 | 12.39 | 11.90 | 9.91 |

*Table 10.* **Comparison of sampling speed.**

| Model | Avg sec per image↓ | FID↓ |
|---|---|---|
| eMIGM-L | 0.165 | 1.77 |
| EDM2-XXL (Karras et al., 2024) | 0.552 | 1.81 |
| EDM2-XXL with guidance interval (Kynkäänniemi et al., 2024) | 0.481 | 1.40 |

**More Evaluation Metrics.** We present additional evaluation metrics on ImageNet $256 \times 256$ in Tab. 8.

**More Mask Schedules.** In this paper, we explored three mask schedules: (1) *Linear*: $\gamma_t = t$; (2) *Cosine*: $\gamma_t = \cos\left(\frac{\pi}{2}(1 - t)\right)$; and (3) *Exp*: $\gamma_t = 1 - \exp(-5t)$. All these schedules are designed to satisfy the approximate boundary conditions $\gamma_0 \approx 0$ and $\gamma_1 \approx 1$. We observed that the exp mask schedule, when used in conjunction with $w(t) = 1$, achieves superior performance compared to other settings.

Furthermore, we developed a log-exp schedule, $\gamma_t = \frac{\log\left(1 + (e^5 - 1) \cdot t\right)}{5}$, which aims to balance mask ratios by reducing extremes in both high and low masking. Following the experimental setup detailed in Fig. 2(b), we present the FID results in Tab. 9. We observed that the log-exp schedule demonstrates improved convergence and performance, thereby validating the benefit of exploring new masking schedules. We leave further investigation of more mask schedules for future work.

**Sampling Speed Comparison with EDM2.** Compared with EDM2's generation network, EDM2's guidance network is relatively small. We therefore conducted additional experiments to compare sampling speeds on a single A100 GPU (batch size 256), with the results presented in Tab. 10. eMIGM-L achieves faster sampling than EDM2-XXL, primarily due to its lower parameter count. Despite requiring a higher NFE, it still maintains competitive performance.

