# OpenReview forum: "Effective and Efficient Masked Image Generation Models"
_ICML.cc/2025/Conference — ICML 2025 poster_

### Official Review · Reviewer_5pkV · 2025-03-13

**Overall Recommendation:** 3

**Summary:**

This paper introduces eMIGM, a unified framework that integrates Masked Image Generation Models and Masked Diffusion Models into a single mathematical formulation. The authors categorize the possible design choices into training and sampling processes to optimize performance and efficiency. By leveraging a time interval strategy for Classifier-Free Guidance (CFG), replacing the fake class token with a mask token, eMIGM achieves comparable performance in image generation while significantly reducing function evaluations (NFEs). Experimental results demonstrate superior efficiency and sample quality on ImageNet 256×256 and 512×512 compared to existing models, including diffusion-based and masked modeling approaches.

**Claims And Evidence:**

The claims made in the paper are mostly supported by clear and convincing evidence:

1. The paper presents a mathematical framework that unifies Masked Image Generation and Masked Diffusion Models, showing that different approaches can be expressed under a generalized loss function. Empirical validation demonstrates that this framework enables systematic exploration of design choices, leading to performance and efficiency improvements.

2. Instead of using a traditional fake class token, the paper proposes replacing it with a mask token in Classifier-Free Guidance (CFG). This modification improves conditional generation by preventing performance degradation caused by fake class tokens, as confirmed through ablation studies.

3. The paper introduces a time interval approach to CFG, where guidance is applied selectively in later sampling steps. This method leads to better FID scores and reduces computational cost, as demonstrated in experiments.

4. (Minor Limitation) Some hyperparameter choices (e.g., mask scheduling functions, weighting strategies) are selected based on empirical tuning rather than theoretical analysis, making it unclear whether the observed improvements generalize beyond the tested configurations.

**Essential References Not Discussed:**

-

**Experimental Designs Or Analyses:**

The paper effectively demonstrates its validity through a well-structured experimental design. To support the possible design choices, Figures 2 and 3 present experimental comparisons of weighting functions, mask schedules, and model architectures, clearly illustrating their impact on performance. To validate image generation performance, Tables 2 and 3 compare eMIGM against a diverse set of generation models, including Diffusion Models, Consistency Models, GANs, ARs, and Masked Models, establishing its performance superiority and scalability, further supported by Figure 4.

However, most design choices are based solely on empirical selection of the best-performing methods, with limited analysis on why these choices lead to improvements. Additionally, while Consistency Models are included in the comparison tables, their differences from eMIGM are not explicitly discussed in the main text. It would be valuable to clarify how eMIGM differs from these models and what makes it superior.

**Methods And Evaluation Criteria:**

The methods and evaluation criteria in the paper are generally appropriate. eMIGM is evaluated on ImageNet 256x256 and 512x512, which are widely used benchmarks for generative models. The paper primarily uses Frechet Inception Distance (FID) to assess image quality and Number of Function Evaluations (NFE) to measure computational efficiency. The proposed time interval strategy is systematically tested through ablation studies and experiments on different mask schedules, providing strong empirical validation. The use of NFE as an efficiency metric is well-justified, as it directly reflects computational cost.

However, a limitation is that the evaluation relies solely on FID and NFE, without incorporating additional metrics such as Inception Score (IS), Recall, or Precision, which could provide a more comprehensive assessment of model performance.

**Other Comments Or Suggestions:**

I believe this paper effectively unifies two approaches into a single framework, systematically categorizing possible design choices to achieve optimal performance. However, rather than introducing entirely new design choices, the paper primarily selects options already used in existing models. While the modifications, such as replacing the fake class token with a mask token in CFG and introducing the time interval strategy, are well-supported by experimental results, the analysis of why these choices improve performance could be further elaborated. Due to this, the paper's overall contribution and novelty may look somewhat limited. I encourage the authors to address these aspects during the rebuttal period.

**Other Strengths And Weaknesses:**

-

**Questions For Authors:**

1. In Figure 2(e), the results appear unstable between epochs 300-400. Do you have experimental results for longer training epochs? While it is clear that CFG with Mask achieves a faster convergence, wouldn’t the final convergence point be the same if trained for a sufficiently long time?
2. Although the experimental results include Consistency Models, the main context does not explore them in detail. What do you think are the differences between eMIGM and Consistency Models?
3. In MaskGIT [1], various mask scheduling options are explored (e.g., cubic, square, square root, logarithmic). This paper considers a more limited selection—was this choice guided by empirical findings or specific constraints? Would exploring additional schedules further improve performance?

**Relation To Broader Scientific Literature:**

This paper is highly relevant to the broader literature on Masked Image Generation and Masked Diffusion Models: It extends prior work on masked generative transformers (MaskGIT) and masked Diffusion models (MDMs) by providing a unified framework​. This paper aligns with research on efficiency improvements in generative models, focusing on reducing NFEs while maintaining sample quality.

**Theoretical Claims:**

All of the mathematical derivations are reasonable, and appropriate references and equivalences are provided, ensuring no issues. Additionally, after reviewing Section 3: Unifying Masked Image Generation and Appendix A: Equivalence of the Masking Strategies of MaskGIT and MDM, no logical errors or inconsistencies were found in the formulation and derivations. The mathematical framework presented in the paper is sound and well-supported by empirical validation.

---

> ### Author Rebuttal · Authors · 2025-04-01
>
> We thank reviewer 5pkV for the interest and acknowledgement of our contributions and the valuable comments. We respond below to your questions and concerns.
> > **Minor Limit** Generalizability of improvements beyond tested configurations unclear
>
> Empirical analysis is a well-established approach in developing complex models. Prior works [a, b] similarly employed ablation studies and targeted experiments to optimize their designs. Through systematic testing of components and hyperparameters, they demonstrated the value of rigorous empirical analysis, even when optimal settings may be context-dependent. Building on this precedent, our study validates specific configurations and provides a framework for systematically exploring and understanding design choices in masked generative models. We believe this methodological contribution offers significant value to the research community.
>
> Besides, eMIGM performs effectively with both VAE (ImageNet 256x256) and DC-AE (ImageNet 512x512) tokenizers despite their distinct latent spaces. This adaptability highlights the transferability of our design principles across different data representations.
>
> [a]“Elucidating the Design Space of Diffusion-Based Generative Models.”
> [b]“Analyzing and Improving the Training Dynamics of Diffusion Models.”
>
> > **Exp1:** More evaluation metrics
>
> We've now evaluated our models using additional metrics (sFID, IS, Precision, and Recall). These comprehensive results are presented in our response to reviewer DbgT's Weakness2 and will be included in the revised paper.
>
> > **Sugg1:** novelty and analysis of the choices of training
>
> We sincerely appreciate your recognition of our unified framework, well-structured experimental design and results.
>
> Regarding your concerns about novelty and analysis, we would like to clarify two key points:
>
> 1. Our hyperparameter tuning follows an empirical approach, aligning with established practices in VAR and MAR. This practical methodology remains valuable as it yields effective, demonstrable results.
>
> 2. Our modifications stem from careful analysis of MDM's behavior. For instance, the time interval strategy was motivated by our observation that MDM's generation process is irreversible, making early-stage guidance less effective - a finding we quantitatively validated in Appendix C. Similarly, replacing fake class tokens with mask tokens in CFG was informed by the fact that MDM has seen more mask tokens during training, making them a more natural choice than the fake class tokens used in diffusion models. We appreciate your suggestion and will expand our analysis of these design choices in the revision.
>
> Our key contribution is a unified framework integrating masked generative transformers and masked diffusion models, advancing masked generative modeling through systematic experimentation and analysis.
>
> > **Q1:** Further training of Figure 2(e)
>
> To reponse this question, we trained standard CFG for 800 epochs. Below we compare its FID scores with mask CFG:
>
> Model|NFE|FID
> -|-|-
> eMIGM-B with standard CFG|16x1.2|2.76
> eMIGM-B with mask CFG|16x1.2|2.79
> eMIGM-B with standard CFG|128x1.35|2.32
> eMIGM-B with mask CFG|128x1.35|2.32
>
> The results show both approaches achieve equivalent final performance, with CFG with Mask converging more rapidly as shown in Fig.2(e) of our paper.
>
> > **Q2:** eMIGM vs Consistency Models
>
> Consistency models and eMIGM employ distinct approaches to generation. Consistency models establish direct mappings from noise to data via consistency constraints, enabling single-step sampling. In contrast, eMIGM utilizes masked prediction, beginning with fully masked images and iteratively revealing content through progressive unmasking. This architectural distinction enables eMIGM to optimally balance generation quality and computational efficiency by modulating function evaluations.
>
> > **Q3:** choice of mask schedule
>
> We explored three mask schedules (linear, cosine, and exponential) with constraints $\gamma_0 \approx 0$ and $\gamma_1 \approx 1$. While linear schedules are common in MDM text generation, we found concave functions like cosine performed better for images due to their information redundancy - higher mask ratios during training provide stronger learning signals. With $w(t)=1$, the exponential schedule slightly outperformed cosine, becoming our default.
>
> Based on your suggestion, we developed a log-exp schedule ($\gamma_t=\frac{\log\left(1 + (e^5 - 1) \cdot t\right)}{5}$) that balances mask ratios by reducing both high and low masking extremes. Using the setup from Figure 2(b) with $w(t)=1$, we present results of FID below.
>
> Epoch|Linear|Cosine|Exp|Log-Exp
> -|-|-|-|-
> 100|38.66|24.99|28.63|25.38
> 200|30.55|16.70|17.97|11.81
> 300|24.55|15.00|11.57|12.48
> 400|24.96|12.39|11.90|9.91
>
> The log-exp schedule shows better convergence and performance, validating the benefits of exploring new masking approaches. We appreciate this insightful suggestion and will incorporate these findings in our revision.

---

> > ### Comment · Reviewer_5pkV · 2025-04-06
> >
> > Thank you for addressing our concerns within the short rebuttal period.
> >
> > Our main concerns were whether the design choices were made with proper analysis, and the issue of limited novelty. However, the authors’ justifications have sufficiently addressed these concerns. In addition, during the rebuttal period, the authors provided additional experimental results with various evaluation metrics and introduced a mask scheduling strategy, which further strengthens the contribution of the work.
> >
> > Given these clarifications and additions, I would like to change my score to Weak accept.

---

> > > ### Author Response · Authors · 2025-04-06
> > >
> > > Dear Reviewer 5pkV,
> > >
> > > We are sincerely grateful for your insightful comments and your decision to update the rating to 'weak accept'. We highly appreciate it.
> > >
> > > Best regards,
> > >
> > > Authors

---

### Official Review · Reviewer_DbgT · 2025-03-14

**Overall Recommendation:** 3

**Summary:**

The paper proposes a unified framework that integrates masked image generation models (e.g., MaskGIT) and masked diffusion models. The authors systematically explore the design space of training and sampling strategies to improve both performance and efficiency. The proposed model, eMIGM, achieves state-of-the-art performance on ImageNet (256×256 and 512×512) while requiring significantly fewer function evaluations (NFEs) compared to continuous diffusion models. Empirical results show that eMIGM outperforms VAR and approaches state-of-the-art diffusion models like REPA, while requiring less than 40% of the NFEs.

**Claims And Evidence:**

The paper makes several claims, and most are well-supported by empirical evidence:

1. Unified framework for masked image modeling and diffusion models.

2. eMIGM achieves better performance than VAR and is competitive with state-of-the-art diffusion models.

3. Proposed sampling strategies (e.g., time interval classifier-free guidance) improve efficiency.

4. Scaling eMIGM improves performance.

However, the claim that eMIGM is "comparable" to continuous diffusion models could be more rigorously defined.

**Essential References Not Discussed:**

Missing reference in masked generative transformer areas[1].

[1] Meissonic: Revitalizing Masked Generative Transformers for Efficient High-Resolution Text-to-Image Synthesis, ICLR 2025.

**Experimental Designs Or Analyses:**

1. Experiments are well-designed, with systematic exploration of training/sampling design choices.

2. Ablation studies (Figures 2 & 3) effectively analyze the contributions of different components.

3. The comparison against state-of-the-art methods is thorough, but it would be beneficial to include additional baselines (e.g., GANs) in more direct comparisons.

**Methods And Evaluation Criteria:**

1. The methods used (masked image modeling, diffusion-based loss, classifier-free guidance) are appropriate for the problem.

2. Evaluation is conducted using ImageNet (256×256 and 512×512), using FID scores.

3. Comparisons with strong baselines (MaskGIT, MAR, diffusion models, VAR) are comprehensive.

4. The choice of FID as the evaluation metric is standard for generative models.

**Other Comments Or Suggestions:**

I will be happy to raise my score if all my concerns are resolved, and vice versa.

**Other Strengths And Weaknesses:**

Strengths:

1. Novel unified framework for masked image modeling and diffusion models.

2. Strong empirical results on ImageNet, with significant efficiency improvements.

3. Comprehensive ablation studies exploring design choices in training and sampling.

4. Well-written and clear methodology, with detailed appendices.

Weaknesses:

1. Limited discussion of failure cases: What types of images does eMIGM struggle with?

2. Lack of qualitative comparisons against diffusion models and VAR beyond FID scores.

**Questions For Authors:**

1. How does eMIGM perform on out-of-distribution datasets? Would the efficiency gains generalize to different image distributions?

2. What are the limitations of eMIGM compared to continuous diffusion models beyond NFEs? Are there failure cases where continuous diffusion models outperform eMIGM?

3. Would a hybrid approach combining eMIGM with diffusion models further improve performance? Could additional modifications bridge the gap to state-of-the-art diffusion performance?

4. How does eMIGM compare in terms of robustness to adversarial perturbations? Could eMIGM be more or less vulnerable compared to diffusion models?

**Relation To Broader Scientific Literature:**

1. The work extends prior efforts in masked image modeling (MaskGIT, MAR) and diffusion models.

2. The proposed unified framework contributes to understanding the connection between masked models and diffusion models.

**Theoretical Claims:**

1. The paper presents a mathematical unification of masked image generation and masked diffusion models.

2. The equivalence between MaskGIT’s loss and MDM’s loss is derived in Appendix A.

3. The mathematical formulation appears correct, though there is nothing new.

---

> ### Author Rebuttal · Authors · 2025-04-01
>
> We thank reviewer DbgT for the interest and acknowledgement of our contributions and the valuable comments. We respond below to your questions and concerns.
>
> > **Claims 1:** Define "comparable" more rigorously
>
> To quantify our comparison: eMIGM achieves an FID of 1.57 on ImageNet 256x256 generation versus REPA's 1.40 - a 12% gap. We consider this competitive since eMIGM requires fewer function evaluations (128×1.4 NFEs vs. REPA's 250×2 NFEs). We will clarify this comparison more precisely in our revised paper.
>
> > **Experiment1:** Include addtional baselines (e.g. GANs)
>
> Our Tables 2 and 3 already include comparisons with GAN baselines (BigGAN and StyleGAN-XL), which we will emphasize more clearly in our revision.
>
> > **References** Missing Meissonic
>
> We appreciate the reference suggestion and will include it in our final paper.
>
> > **Weakness1:** Limited discussion of failure cases: What types of images does eMIGM struggle with?
>
> While we have not observed distinct failure patterns, eMIGM does experience occasional generation failures, as do other methods like MAR and VAR. We will include representative failure cases in our revision.
>
> > **Weakness2:** Qualitative comparisons beyond FID
>
> We provide qualitative visual comparisons with diffusion models and VAR at [link](https://anonymous.4open.science/r/icml-rebuttal-6388/icml_link.pdf) and quantitative evaluations using sFID, IS, Precision, and Recall (see Table below), both of which will be included in our revision.
>
> Method|NFE|FID↓|sFID↓|IS↑|Precision↑|Recall↑
> -|-|-|-|-|-|-
> VAR-d30|10×2|1.92|-|323.1|0.82|0.59
> -|-|-|-|-|-|-
> REPA|250×2|1.42|4.70|305.7|0.80|0.65
> -|-|-|-|-|-|-
> eMIGM-XS|16x1.2|4.23|5.74|218.63|0.79|0.50
> eMIGM-S|16x1.2|3.44|5.31|244.16|0.80|0.53
> eMIGM-B|16x1.2|2.79|5.20|284.62|0.82|0.54
> eMIGM-L|16x1.2|2.22|4.80|291.62|0.80|0.59
> eMIGM-H|16x1.2|2.02|4.66|299.36|0.80|0.60
> -|-|-|-|-|-|-
> eMIGM-XS|128x1.4|3.62|5.47|224.91|0.80|0.51
> eMIGM-S|128x1.4|2.87|5.53|254.48|0.80|0.54
> eMIGM-B|128x1.35|2.32|4.63|278.97|0.81|0.57
> eMIGM-L|128x1.4|1.72|4.63|304.16|0.80|0.60
> eMIGM-H|128x1.4|1.57|4.68|305.99|0.80|0.63
>
> > **Q1:** How does eMIGM perform on out-of-distribution datasets and different image distributions?
>
> eMIGM demonstrates strong adaptability across data representations, performing effectively with both VAE (256x256) and DC-AE (512x512) tokenizers despite their different latent spaces, showing our efficiency improvements generalize across resolutions and distributions.
>
> Regarding out-of-distribution, while important, our research focuses primarily on improving the efficiency and quality of the generative models themselves, following the established research direction of prior works like MAR and VAR. These foundational works concentrate on in-distribution generation, which remains our focus for high-quality image synthesis.
>
> > **Q2:** What limitations does eMIGM have compared to continuous diffusion models? When do diffusion models outperform eMIGM?
>
> Compared to continuous diffusion models, eMIGM has limitations in zero-shot classification [a], ultimate generation quality with sufficient NFEs [b], and applications in video/audio synthesis. However, we believe masked generative models can narrow this gap as they evolve. We will discuss these limitations comprehensively in our revision.
>
> [a]“Your Diffusion Model is Secretly a Zero-Shot Classifier.”
> [b]“Inference-Time Scaling for Diffusion Models beyond Scaling Denoising Steps.”
>
> > **Q3:** Hybrid approach with diffusion models and modifications?
>
> A hybrid approach combining eMIGM with diffusion models could enhance performance. Recent works [c,d] show that integrating autoregressive and diffusion models achieves superior generation capabilities compared to pure continuous diffusion models. Additionally, simple modifications could bridge remaining performance gaps, such as adopting rectified flow [e] for the diffusion head's training objective or following REPA [f] by aligning eMIGM's intermediate Transformer outputs with pretrained DINOv2 encoder features. We will incorporate this discussion in our revised paper.
>
> [c]“Efficient Visual Generation with Hybrid Autoregressive Transformer”
> [d]“Diffusion Transformer Autoregressive Modeling for Speech Generation”
> [e]“Learning to Generate and Transfer Data with Rectified Flow”
> [f]“Training Diffusion Transformers Is Easier Than You Think”
>
> > **Q4:** How does eMIGM's robustness to adversarial perturbations compare to diffusion models?
>
> We appreciate your insightful question regarding adversarial robustness. While we acknowledge the critical importance of this aspect in the broader context of generative modeling, our current research primarily focuses on enhancing the efficiency and quality of the generative models themselves. We acknowledge that our expertise in adversarial perturbations is limited. Currently, we have not performed a assessment of eMIGM's robustness against such perturbations. We will discuss this important aspect in our revised paper.

---

### Official Review · Reviewer_qvwy · 2025-03-16

**Overall Recommendation:** 3

**Summary:**

This paper provides a comprehensive study of masked diffusion models for visual generation, covering training, sampling, and architectural designs with extensive experiments. In other words, this paper investigates how to make a good MDM with regard to training and sampling settings through empirical evidences.

**Claims And Evidence:**

The claims made in the submission are supported by clear and convincing empirical evidence.

**Essential References Not Discussed:**

There are no highly-related work missing in this paper in my view.

**Experimental Designs Or Analyses:**

I checked the soundness and validity of experimental designs and analyses. There are some concerns.

1. I don't quite understand the paragraph "CFG with Mask" in line 246 and the difference between Mask CFG and fake class CFG is hard to distinguish. The authors should put more details on what is the special mask token here.

2. The network architectural details come too late in experiments. I wasn't aware of the design of continuous targets in eMIGM framework before Section 5.2, as the conventional masked based generative models typically use discrete targets to perform classification. So there comes the problem, what is the baseline performance of the discrete variant of eMIGM?

3. I don't think some analysis, especially when compared to diffusion model steps, make sense. Since eMIGM use diffusion loss, there are multi-step diffusion steps in a single mask step.

**Methods And Evaluation Criteria:**

The proposed methods and evaluation criteria make sense for the problem.

**Other Comments Or Suggestions:**

See Questions for Authors

**Other Strengths And Weaknesses:**

Overall, this is a borderline paper with both strengths and weaknesses. It includes extensive experiments that thoroughly explore the training and sampling space of masked diffusion models, which should be highly appreciated. On the other side, this paper lacks a clear goal or motivation to unify the masked diffusion models rather than performance. Additionally, despite extensive trial and error across many experiments, the largest eMIGM variant performs only on par with MAR-H, which raises concerns about the effectiveness of the proposed method, or the necessity of the unified masked diffusion modeling framework.

**Questions For Authors:**

1. What makes the abbreviation "eMIGM"? Is it something like "EDM" [1]?

2. When discussing mask-based generative models, the authors should first clarify whether they are using discrete or continuous models. Also, the authors should justify why they are using continuous models like MAR instead of MaskGIT pipeline.

[1] Elucidating the Design Space of Diffusion-Based Generative Models

**Relation To Broader Scientific Literature:**

The key contributions of this paper is highly related to the MAR and MaskGIT, especially in terms of masking mechanism and diffusion loss head.

**Theoretical Claims:**

There are almost no theoretical claims in this submission except for the equivalence of MaskGIT and MDM masking strategies. I checked the correctness of proof in Appendix A.

---

> ### Author Rebuttal · Authors · 2025-04-01
>
> > **Experimental Design1:** The claim of "CFG with Mask" and "fake class CFG"
>
> Thank you for your question. In standard Classifier-Free Guidance (CFG) used in diffusion models and MAR, training involves occasionally replacing the class label with a dedicated 'fake class' token, which is distinct from the 'mask' token used for image patches. Our "CFG with Mask" approach instead uses the existing 'mask' token to replace class labels during training, eliminating the need for a separate 'fake class' token. We will clarify this distinction in the final paper.
>
> > **Experimental Design2:** Network architectural details come too late and What is the baseline performance of the discrete variant of eMIGM?
>
> Thank you for your valuable suggestion. We appreciate your feedback and will incorporate more detailed network architectural information in an earlier section of our paper to enhance clarity. As briefly mentioned at the beginning of Section 4, our experiments exclusively focus on continuous masked-based generative models to mitigate the information loss associated with discrete tokenizers. Consequently, we did not develop a discrete variant of eMIGM, particularly since prior work has demonstrated that discrete variants of MAR perform significantly worse than their continuous counterparts. We will ensure all experimental details are presented more clearly in the final version of our paper.
>
> > **Experimental Design3:** The model steps of eMIGM.
>
> We sincerely appreciate your valuable feedback. In response, we conducted an experiment on 512x512 image generation using eMIGM-L with NFE=64x1.25 (as presented in Table 3 of our paper). In this configuration, the diffusion model requires 14 sampling steps. To determine the additional computational cost of the diffusion model beyond the main transformer's NFEs, we measured the sampling speed with different diffusion steps. Our measurements on a single A100 GPU with batch size 256 show that the diffusion model introduces approximately 14% additional computational overhead beyond the main transformer's NFE requirements. Thank you for raising this concern about NFE comparisons between eMIGM and diffusion models. While eMIGM does include diffusion steps within each masked step, since transformer forward passes remain the primary bottleneck, NFE continues to be a valid efficiency metric. We will address this point more thoroughly in the revised paper, ensuring clarity and precision in our presentation.
>
>
> > **Weakness1:** The effectiveness of the proposed method, or the necessity of the unified masked diffusion modeling framework.
> We appreciate your valuable feedback.
>
> Regarding the necessity of the unified masked diffusion modeling framework, it serves two primary purposes. First, it enhances the theoretical understanding of masked generation and simplifies comparisons between different methods (e.g., MAR vs. MaskGIT). Second, it facilitates a systematic exploration of the design space and allows for the integration of techniques from related areas. For example, advancements in diffusion models can be readily incorporated into our diffusion head training, helping to advance the state-of-the-art in masked image generation.
>
> Regarding the effectiveness of eMIGM, our comprehensive experimental analysis demonstrates key advantages:
>
> 1.  Through systematic exploration of the training and sampling design space, we identified critical factors influencing model performance and efficiency. These findings provide actionable insights for developing future masked generative models.
> 2.  The proposed time-interval strategy for classifier-free guidance (CFG) empirically improves sampling efficiency while maintaining generation quality.
>
> These improvements are substantiated by direct comparisons with MAR baselines. For ImageNet 512×512 generation, eMIGM achieves superior computational efficiency, requiring only 64×1.25 NFEs compared to MAR's 256×2 NFEs while maintaining competitive FID scores.
>
> > **Q1:** what makes the abbreviation "eMIGM"?
>
> Thank you for your question. The abbreviation "eMIGM" stands for "Effective and Efficient Masked Image Generation Models.", where the prefix 'e' denotes both effectiveness and efficiency.
>
> > **Q2:** Why using continuous models like MAR instead of MaskGIT pipeline.
>
> Thank you for this insightful question. We will clarify that our work focuses exclusively on continuous masked-based generative models. As briefly noted in Section 4, this choice stems from the need to avoid information loss inherent in discrete tokenizers. Previous research has consistently shown that discrete variants of MAR yield substantially inferior performance compared to continuous implementations. We will emphasize this choice more clearly in the final version of our paper and apologize for any previous lack of clarity.

---

### Official Review · Reviewer_Bt5L · 2025-03-17

**Overall Recommendation:** 3

**Summary:**

The paper presents eMIGM, a novel model for effective and efficient masked image generation. It unifies masked image generation models and masked diffusion models within a single framework, exploring the design space of training and sampling to identify key factors impacting performance and efficiency. The model demonstrates strong performance on ImageNet generation, outperforming seminal models like VAR and achieving results comparable to state-of-the-art continuous diffusion models with significantly fewer NFEs. Key contributions include a unified formulation for exploring the design space, a time interval strategy for classifier-free guidance, surpassing state-of-the-art diffusion models on ImageNet 512×512 with fewer NFEs, and demonstrating scalability benefits.

**Claims And Evidence:**

The claims made in the submission are generally supported by clear and convincing evidence. The authors provide extensive experimental results on ImageNet 256×256 and 512×512, comparing eMIGM with state-of-the-art generative models. They demonstrate that eMIGM achieves lower FID scores with fewer NFEs and parameters compared to models like VAR and even outperforms some diffusion models. The ablation studies on different components (mask schedule, weighting function, model architecture, time truncation, CFG with mask) provide strong evidence for the design choices made. However, some claims could be further strengthened by additional analyses. For instance, while the authors claim that larger models are more training and sampling efficient, a more detailed analysis of the computational resources required for different model sizes and the trade-offs between model size and performance would provide more comprehensive support.

**Essential References Not Discussed:**

[1] Hart: Efficient visual generation with hybrid autoregressive transformer, ICLR 2025.
[2] Meissonic: Revitalizing Masked Generative Transformers for Efficient High-Resolution Text-to-Image Synthesis, ICLR 2025.
[3] Bag of Design Choices for Inference of High-Resolution Masked Generative Transformer
[4] AdaNAT: Exploring Adaptive Policy for Token-Based Image Generation, ECCV 2024

**Experimental Designs Or Analyses:**

The experimental designs and analyses are sound and valid. The authors conduct extensive experiments on ImageNet datasets, comparing eMIGM with a wide range of generative models, including diffusion models, autoregressive models, GANs, and other masked models.

**Methods And Evaluation Criteria:**

The proposed methods and evaluation criteria are appropriate for the problem of masked image generation. The unified framework integrating masked image modeling and masked diffusion models makes sense, as it allows for a systematic exploration of the design space. The choice of Fréchet Inception Distance (FID) as the evaluation metric is standard in the field and suitable for comparing image generation quality. The experimental designs, including comparisons with various state-of-the-art models and ablation studies, are well-structured to validate the effectiveness and efficiency of eMIGM.

**Other Comments Or Suggestions:**

NAN

**Other Strengths And Weaknesses:**

Strengths:
The paper presents a novel unified framework that integrates masked image generation and diffusion models, offering a systematic approach to exploring the design space.
Extensive experiments demonstrate the effectiveness and efficiency of eMIGM, showing strong performance on ImageNet with fewer computational resources compared to state-of-the-art models.
The introduction of the time interval strategy for classifier-free guidance is a valuable contribution that improves sampling efficiency.
The ablation studies provide valuable insights into the impact of different design choices, helping to guide future research in this area.
Weaknesses:
While the paper shows strong results on ImageNet, the generalizability to other datasets and domains could be further explored.
The paper could benefit from a more detailed discussion of the practical implications of using eMIGM

**Questions For Authors:**

How does eMIGM handle different image resolutions beyond those tested in the paper (256×256 and 512×512)? Are there any limitations or adjustments needed when applying the model to higher or lower resolution images?
What are the main computational bottlenecks when scaling up eMIGM models, and are there any plans to explore more efficient architectural designs?
Could you discuss the potential applications of eMIGM beyond image generation?

**Relation To Broader Scientific Literature:**

The key contributions of the paper are well-related to the broader scientific literature. The work builds upon and advances previous research in masked image modeling (e.g., MaskGIT, MAR) and masked diffusion models, integrating their strengths within a unified framework. It also connects to the extensive literature on diffusion models, autoregressive models, GANs, and other generative models, positioning eMIGM as a competitive alternative.

**Theoretical Claims:**

the conceptual framework and derivations related to unifying masked image generation and diffusion models seem logically consistent based on the explanations provided

---

> ### Author Rebuttal · Authors · 2025-04-01
>
> > **Claims:** Some claims could be further strengthened by additional analyses.
>
> We appreciate your feedback regarding our efficiency claims.
>
> Our analysis of training efficiency examined the relationship between training FLOPs and FID scores. As shown in Fig.4(b), larger eMIGM models achieve better FID scores at equivalent FLOP budgets—for example, eMIGM-L outperforms eMIGM-B when both consume approximately 10^20 FLOPs, confirming superior training efficiency of larger models.
>
> Regarding sampling efficiency, we evaluated the inference speed-FID trade-off (Fig.4(c)), with measurements conducted on an A100 GPU using batch size 256. Results demonstrate that larger eMIGM consistently deliver better FID scores than smaller models at comparable inference speeds.
>
> We will follow your suggestion to enhance the clarity of these findings in our revised paper.
>
> > **References:** Some references are not discussed.
>
> Thank you for suggesting these reference. We will incorporate a discussion of them into the revised paper.
>
> > **Weak1:** The generalizability to other datasets and domains could be further explored.
>
> We appreciate your feedback regarding the generalizability of eMIGM. eMIGM performs effectively with both VAE (ImageNet 256x256) and DC-AE (ImageNet 512x512) tokenizers despite their distinct latent spaces. This adaptability highlights the transferability of our design principles across different data representations. We plan to explore eMIGM's application across diverse datasets and domains in future work.
>
> > **Weak2:** The paper could benefit from a more detailed discussion of the practical implications of using eMIGM
>
> We appreciate this valuable suggestion. The practical implications of eMIGM are significant and multifaceted. Our model achieves higher-quality sample generation with fewer sampling steps (NFE), offering substantial advantages in computational resource utilization. Based on these efficiency gains, we believe eMIGM has strong potential for effective application across diverse domains currently served by diffusion models, including text to image and video generation. Besides, we systematically identify better design choices in masked generative modeling, offering practical guidance for developing efficient models. Our framework demonstrates how empirical analysis can effectively guide architectural decisions in this domain.
>
> > **Q1:** Are there any limitations or adjustments needed when applying the model to higher or lower resolution images?
>
> Thank you for this question. When applying eMIGM to different resolutions, we find that the model architecture is inherently resolution-agnostic. The transformer backbone can process different sequence lengths without architectural modifications, allowing for flexible adaptation to various image resolutions. Our experiments on both 256×256 and 512×512 resolutions demonstrate this capability. The primary adjustment needed is simply retraining the model on the target resolution data, as is standard practice with most generative models.
>
> > **Q2:** What are the main computational bottlenecks when scaling up eMIGM models
>
> Thank you for this important question. In our current implementation and experiments with eMIGM, we have not yet encountered significant computational bottlenecks that would require specialized optimization techniques. The model scales effectively with our available computational resources.
>
> > **Q3:** are there any plans to explore more efficient architectural designs?
>
> We appreciate this valuable question. For future efficient architectural designs, we are exploring two promising directions: (1) U-ViT architecture, which achieves comparable performance to DiT on ImageNet 256x256 while using only ~30% of training FLOPs through its U-shaped design; and (2) REPA-inspired alignment between eMIGM's intermediate Transformer outputs and pretrained DINOv2 encoder features. Our preliminary experiments with the latter approach showed slight improvements when aligning earlier layers (4th), while later layers (18th) degraded performance. Though current benefits are marginal, more systematic investigation could yield substantial efficiency gains in future work.
>
> > **Q4:** Could you discuss the potential applications of eMIGM beyond image generation?
>
> We appreciate this important question about eMIGM's broader applications. The model's core mechanism of predicting masked content from surrounding context naturally extends to various image manipulation tasks including inpainting (treating missing regions as masks), conditional editing (regenerating masked objects based on specified conditions), and outpainting (predicting exterior content around a central image). Beyond images, we believe eMIGM could be adapted for text to image, video, and audio generation, though this would require carefully designed input formats and potential architectural modifications to accommodate these diverse data modalities.

---

### Official Review · Reviewer_7cuF · 2025-03-17

**Overall Recommendation:** 4

**Summary:**

This work explores masked diffusion and image modeling through a unified framework, systematically analyzing several key design choices in this domain. Within this framework, the authors ablate masking schedules, loss weighting, and sampling strategies, leading to improvements over existing standards in each area. Building on these insights, they propose eMIGM, an improved masked diffusion modeling method that rivals state-of-the-art continuous diffusion models like EDM2. Experimental results demonstrate that eMIGM achieves competitive FID scores on ImageNet at 256$\times$256 and 512$\times$512 resolutions while requiring fewer NFEs than most diffusion models.

**Claims And Evidence:**

The paper’s main claims regarding performance and efficiency are well-supported by the experimental results. However, some claims should be rephrased to better align with common practices in prior works. Specifically, using a weight schedule for guidance is a well-established technique for enhancing the diversity of CFG, as proposed in several recent works [1, 2, 3]. Additionally, the unsupervised guidance method was introduced in [4] and is closely related to [5]. To ensure clarity and accuracy, the authors should revise certain sections to avoid presenting these established methods as novel contributions of this work.

[1] Sadat S, Buhmann J, Bradley D, Hilliges O, Weber RM. CADS: Unleashing the diversity of diffusion models through condition-annealed sampling. arXiv preprint arXiv:2310.17347. 2023 Oct 26.

[2] Kynkäänniemi T, Aittala M, Karras T, Laine S, Aila T, Lehtinen J. Applying guidance in a limited interval improves sample and distribution quality in diffusion models. arXiv preprint arXiv:2404.07724. 2024 Apr 11.

[3] Wang X, Dufour N, Andreou N, Cani MP, Abrevaya VF, Picard D, Kalogeiton V. Analysis of classifier-free guidance weight schedulers. arXiv preprint arXiv:2404.13040. 2024 Apr 19.

[4] Nie S, Zhu F, Du C, Pang T, Liu Q, Zeng G, Lin M, Li C. Scaling up Masked Diffusion Models on Text. arXiv preprint arXiv:2410.18514. 2024 Oct 24.

[5] Karras T, Aittala M, Kynkäänniemi T, Lehtinen J, Aila T, Laine S. Guiding a diffusion model with a bad version of itself. Advances in Neural Information Processing Systems. 2024 Dec 16;37:52996-3021.

**Essential References Not Discussed:**

Please refer to the Claims and Evidence section for further details.

**Experimental Designs Or Analyses:**

The experiments are well-designed and clearly show how different components affect the performance of eMIGM.

**Methods And Evaluation Criteria:**

The authors use well-established benchmarks to evaluate the performance of eMIGM.

**Other Comments Or Suggestions:**

- The discussion on the mask schedule suggests that the cosine schedule is optimal, yet the authors ultimately use the exponential schedule. This should be clarified—one way to address this is to discuss the weighting strategy and masking schedule together to explain why the exponential schedule with $w(t) = 1$ is the final choice.
- If the diffusion model is used for clean data prediction similar to MAR, it is unclear how much additional sampling cost is introduced by the diffusion model beyond the NFEs required for the main transformer.
- Figure 4 is misplaced.

**Other Strengths And Weaknesses:**

### **Strengths**
- The paper is well-written, with clear and easy-to-follow explanations.
- The experiments are thorough, with a structured, step-by-step introduction that enhances readability and understanding.
- Improving the performance of discrete diffusion models for image generation is an important and timely research direction.

### **Weaknesses**
- Some prior works are not adequately discussed, which affects the accuracy of certain novelty claims.
- The number of NFEs used by the model remains relatively high (e.g., exceeding 64), which slightly contradicts the paper's efficiency claims., although the method still performs well with lower NFEs.
- The comparisons in Tables 2 and 3 are not entirely fair, as most reported FIDs are calculated using constant CFG. For instance, EDM2 with Guidance Interval achieves an FID of 1.40, whereas the reported value is 1.81. This discrepancy should be addressed for a more accurate comparison.
- The NFE comparison with EDM2 is not entirely representative, as EDM2's guidance network is relatively small. Consequently, the NFEs for the conditional and unconditional parts do not have the same computational cost. A more appropriate metric for comparing the sampling speed of eMIGM with EDM2 would be generation throughput (images/sec).

**Questions For Authors:**

1) Have you experimented with other weighting and masking functions, such as sigmoid masking or the log-normal weighting from EDM?

2) How does the method perform when using fewer than 16 sampling steps?

3) Could the diffusion component for data prediction benefit from classifier-free guidance instead of temperature sampling?

4) Does the model still receive the class condition as input when using the CFG mask token for training?

**Relation To Broader Scientific Literature:**

The authors discuss most of the relevant prior work in the main text. Masked diffusion modeling is a novel technique with significant potential for generative modeling, yet it remains underexplored. Since most existing methods in this domain have been developed for language modeling, the authors' contribution to advancing performant MDMs for image generation is noteworthy. As such, the paper is well-positioned within the current literature on masked diffusion models. Moreover, the proposed unified framework can facilitate more systematic research into the performance of these models. That said, some claims presented as new contributions require greater clarity. Please refer to the Claims and Evidence section for further details.

**Theoretical Claims:**

There is no major theoretical claim in the paper.

---

> ### Author Rebuttal · Authors · 2025-04-01
>
> We thank reviewer 7cuF for the interest and acknowledgement of our contributions and the valuable comments. We respond below to your questions and concerns.
>
> > **Claim1:** using a weight schedule for guidance
>
> Compared to existing work, our approach is motivated by MDMs' unique irreversible token generation constraint. We found that early strong guidance restricts generation diversity and increases FID scores. We will revise our paper to properly acknowledge prior work and clarify our specific contributions to guidance strategies for MDMs.
>
> > **Claim2 and Weak1**: unsupervised guidance method
>
> We will clarify that we adapt the unsupervised CFG from text generation [4] to image generation. For clarity, we rename it to "CFG with Mask" to better reflect our focus on masked image generation. Besides, we will add a discussion of [5] in the final version.
>
> > **Weak2:** number of NFEs remains relatively high
>
> Our work focuses on improving masked image modeling efficiency. For example, in 512x512 generation, eMIGM-L matches MAR's performance with fewer NFEs (64x1.25 vs 256x2). We believe future work applying distillation could further improve efficiency.
>
> > **Weak3:** Tables 2 and 3 are not entirely fair
>
> While we previously cited results from the original EDM2 paper, we will update our comparison to include EDM2 with Guidance Interval [2].
>
> > **Weak4:** NFE comparison with EDM2
>
> We conducted additional experiments comparing sampling speeds on a single A100 GPU (batch size 256). As shown below, eMIGM achieves faster sampling than EDM2 while maintaining competitive FID.
>
> Model|Avg sec per image↓|FID↓
> -|-|-
> eMIGM-L|0.165|1.77
> EDM2-XXL|0.552|1.81
> EDM2-XXL with interval|0.481|1.40
>
> > **Sugg1:** More discuss about weighting strategy and masking schedule
>
> Our experiments showed that weighting functions significantly affect noise schedule performance. Using $w(t)=\frac{\gamma_t'}{\gamma_t}$ led to unstable training, especially with the exp schedule. Switching to $w(t)=1$ stabilized training across all schedules and improved performance, with the exp schedule yielding the best results. We adopted this combination as our default and will clarify this relationship.
>
> > **Sugg2:** About additional sampling cost
>
> In our 512x512 image generation experiments with eMIGM-L (NFE=64x1.25), the diffusion model need 14 sampling steps using DPM-Solver, introducing approximately 14% additional sampling speed on a single A100 GPU (batch size 256). This is more efficient than MAR, which requires 100 diffusion steps. As shown in Response to W4, eMIGM achieves faster sampling (0.165s vs 0.552s per image) while maintaining competitive FID scores compared to EDM2. We will include a efficiency comparison table in our final version.
>
> > **Sugg3:** Fig.4 is misplaced
>
> We will relocate Fig.4 to be positioned near Section 6.1.
>
> > **Q1:** Experimented with other weighting and masking functions
>
> Regarding to the mask schedules, we explored three mask schedules (linear, cosine, and exponential) constrained by $\gamma_0 \approx 0$ and $\gamma_1 \approx 1$. In response to your question, we introduce a log-exp schedule that balances mask ratios by reducing extreme cases of both high and low masking in exp schedule. The schedule is defined as: $\gamma_t=\frac{\log\left(1 + (e^5 - 1) \cdot t\right)}{5}$. Following the experimental setup in Figure 2(b) with $w(t)=1$, we present comparative FID in the table below.
> Epoch|Linear|Cosine|Exp|Log-Exp
> -|-|-|-|-
> 100|38.66|24.99|28.63|25.38
> 200|30.55|16.70|17.97|11.81
> 300|24.55|15.00|11.57|12.48
> 400|24.96|12.39|11.90|9.91
>
> The log-exp schedule shows better convergence and performance, validating the benefits of exploring new masking schedule.
>
> Regarding weighting functions, we explored two primary options: $w(t) = \frac{\gamma_t^\prime}{\gamma_t}$ and $w(t) = 1$. The log-normal weighting from EDM isn't directly transferable to our framework. Exploring alternative weighting functions remains promising for future work.
>
> > **Q2:** Performance with <16 sampling steps
>
> We evaluated eMIGM's performance with fewer sampling steps on ImageNet 256×256, with results shown in the table below:
> Method|NFE|FID
> -|-|-
> eMIGM-XS|8x1.2|5.19
> eMIGM-S|8x1.2|4.69
> eMIGM-B|8x1.2|3.77
> eMIGM-L|8x1.2|3.30
> eMIGM-H|8x1.2|3.07
>
> > **Q3:** Could diffusion component benefit from cfg
>
> Our diffusion component already implements classifier-free guidance. During sampling, our model processes both class and mask token inputs to generate $z_c$ and $z_u$ outputs. CFG guides generation toward class-conditioned results using the formula $\epsilon=\epsilon_\theta(x_t|t,z_u)+\omega\cdot(\epsilon_\theta(x_t|t,z_c)-\epsilon_\theta(x_t|t,z_u))$, where $\omega$ is the guidance scale. We will clarify this in our final paper.
>
> > **Q4:** model receive the class condition as input
>
> The model still receives class conditions during training, but we randomly replace the true label with a mask token at a fixed probability. We will clarify this in our revised paper.

---

> > ### Comment · Reviewer_7cuF · 2025-04-07
> >
> > I would like to thank the authors for answering my questions in the rebuttal. I also have the following remaining questions:
> >
> > 1. If the model uses a mask token as input, how is this different from learning an unconditional model in classifier-free guidance (CFG) by introducing an additional empty class?
> >
> > 2. Do you have comparisons of eMIGM-H using 8 sampling steps with other diffusion-based models operating at similarly low step counts? While this is not strictly necessary for the rebuttal, given the strong performance of eMIGM-H in such settings, it could further enhance the contributions of the paper.
> >
> > 3. Similarly, since the Log-Exp schedule appears to outperform other scheduling methods, I’m curious what the final performance would be when combined with this masking strategy.
> >
> > Overall, I believe the paper is well-written, and the core contributions are interesting. Although some components build on prior work, the unification of different approaches could be valuable for future research in masked image modeling—especially if the authors release the code. Therefore, I would like to increase my score to Accept.
> >
> > **Minor comment**: The fact that early strong guidance negatively impacts diversity has been previously noted in [1, 2]. Please ensure that related prior work is appropriately discussed and cited in the final version of the paper.
> >
> > [1] Sadat S, Buhmann J, Bradley D, Hilliges O, Weber RM. CADS: Unleashing the diversity of diffusion models through condition-annealed sampling. arXiv preprint arXiv:2310.17347. 2023 Oct 26.
> >
> > [2] Kynkäänniemi T, Aittala M, Karras T, Laine S, Aila T, Lehtinen J. Applying guidance in a limited interval improves sample and distribution quality in diffusion models. arXiv preprint arXiv:2404.07724. 2024 Apr 11.

---

> > > ### Author Response · Authors · 2025-04-09
> > >
> > > We thank Reviewer 7cuF for acknowledging our contributions since the beginning. We are glad that the vast majority of the concerns have been addressed. We respond below to your remaining questions.
> > >
> > > > **Q1:** If the model uses a mask token as input, how is this different from learning an unconditional model in classifier-free guidance (CFG) by introducing an additional empty class?
> > >
> > > When using mask tokens as input, the training of unconditional models is fundamentally equivalent since mask tokens introduce no additional information. However, implementation-wise, using an additional empty class for unconditional model training optimizes both the mask token and empty class as trainable tensors, whereas using mask tokens as input only optimizes the mask token itself.
> > >
> > > > **Q2:** Do you have comparisons of eMIGM-H using 8 sampling steps with other diffusion-based models operating at similarly low step counts?
> > >
> > > We conducted a comparative analysis of eMIGM-H against other diffusion-based models at low step counts on ImageNet 256×256, with results shown below:
> > >
> > > Method|NFE|FID|IS
> > > -|-|-|-
> > > eMIGM-H|8x1.2|3.07|299.4
> > > eMIGM-H|16x1.2|2.02|299.4
> > > -|-|-|-
> > > DiT-XL/2|8x2|63.9|53.8
> > > DiT-XL/2|16x2|19.5|136.9
> > > -|-|-|-
> > > REPA|8x1.7|100.19|22.7
> > > REPA|16x1.7|15.29|170.6
> > >
> > > Using official implementations of DiT and REPA with modified sampling steps, we observe that eMIGM-H demonstrates superior performance at low NFEs. Specifically, with 8x1.2 steps, eMIGM-H achieves an FID of 3.07 and IS of 299.4, significantly outperforming both DiT-XL/2 (FID: 63.9, IS: 53.8) and REPA (FID: 100.19, IS: 22.7) under comparable settings. These results highlight eMIGM-H's exceptional efficiency and generation quality at low step counts. We will follow your suggestion to incorporate these findings in our revised paper.
> > >
> > > > **Q3:** what the final performance would be when combined with the Log-Exp mask schedule.
> > >
> > > We sincerely appreciate your valuable question. In response, we are currently conducting this experiment. However, due to time constraints, the experiment requires approximately two more days to complete, while the reviewer-author rebuttal period concludes in just a few hours. Therefore, we will include these results in our revised paper.
> > >
> > > > **Minor Comment:** Please ensure that related prior work [1,2] is appropriately discussed and cited in the final version of the paper.
> > >
> > > We sincerely appreciate you bringing these related prior works to our attention. We commit to thoroughly discussing these works and properly citing them in the final version of our paper. Thank you again for your valuable feedback.
> > >
> > > > **Acknowledgment of Positive Feedback**
> > >
> > > We sincerely thank the reviewer for their positive feedback and for recognizing the value of our work. We are particularly grateful for the decision to increase the score to Accept, which is a strong endorsement of our research contributions. We appreciate the reviewer's thoughtful consideration throughout the review process and their constructive suggestions that have helped improve our paper. We look forward to incorporating all feedback in our final version.

---

### Decision · Program_Chairs · 2025-05-01

**Decision:**

Accept (poster)

**Comment:**

This work delves into masked diffusion and image modeling via a unified framework. It conducts a systematic analysis of several crucial design aspects in this field. In the framework, the authors conduct ablation studies on masking schedules, loss weighting, and sampling strategies, achieving enhancements compared to current standards in each of these aspects. Leveraging the insights gained, they introduce eMIGM, an advanced masked diffusion modeling approach that can compete with top-notch continuous diffusion models such as EDM2.

All reviewers (five) recommend acceptance of this work with all the problems sovled. Thus, the AC recommend acceptance with new masked diffusion framework and good performance.